# Finite-Time Analysis for Conflict-Avoidant Multi-Task Reinforcement Learning

## Abstract

Multi-task reinforcement learning (MTRL) has shown great promise in many real-world applications. Existing MTRL algorithms often aim to learn a policy that optimizes individual objective functions simultaneously with a given prior preference (or weights) on different tasks. However, these methods often suffer from the issue of *gradient conflict* such that the tasks with larger gradients dominate the update direction, resulting in a performance degeneration on other tasks. In this paper, we develop a novel dynamic weighting multi-task actor-critic algorithm (MTAC) under two options of sub-procedures named as CA and FC in task weight updates. MTAC-CA aims to find a conflict-avoidant (CA) update direction that maximizes the minimum value improvement among tasks, and MTAC-FC targets at a much faster convergence rate. We provide a comprehensive finite-time convergence analysis for both algorithms. We show that MTAC-CA can find a $\epsilon + \epsilon_{\text{app}}$-accurate Pareto stationary policy using $\mathcal{O}(\epsilon^{-5})$ samples, while ensuring a small $\epsilon + \sqrt{\epsilon_{\text{app}}}$-level CA distance (defined as the distance to the CA direction), where $\epsilon_{\text{app}}$ is the function approximation error. The analysis also shows that MTAC-FC improves the sample complexity to $\mathcal{O}(\epsilon^{-3})$, but with a constant-level CA distance. Our experiments on MT10 demonstrate the improved performance of our algorithms over existing MTRL methods with fixed preference.

## 1 Introduction

Reinforcement learning (RL) has made much progress in a variety of applications, such as autonomous driving, robotics manipulation, and financial trades Deng et al. (2016); Sallab et al. (2017); Gu et al. (2017). Though the progress is significant, much of the current work is restricted to learning the policy for one task Mülling et al. (2013); Andrychowicz et al. (2020). However, in practice, the vanilla RL polices often suffers from performance degradation when learning multiple tasks in a multi-task setting. To deal with these challenges, various multi-task reinforcement learning (MTRL) approaches have been proposed to learn a single policy or multiple policies that maximize various objective functions simultaneously. In this paper, we focus on single-policy MTRL approaches because of their better efficiency. On the other side, the multi-policy method allows each task to have its own policy, which requires high memory and computational cost. The objective is to solve the following MTRL problem:

$$\max_{\pi} \mathbf{J}(\pi) := (J^1(\pi), J^2(\pi), ..., J^K(\pi)), \tag{1}$$

where $K$ is the total number of tasks and $J^k(\pi)$ is the objective function of task $k \in [K]$ given the policy $\pi$. Typically, existing single-policy MTRL methods aim to find the optimal policy with the given preference (i.e., the weights over tasks) ). For example, Mannor & Shimkin (2001) developed a MTRL algorithm considering the average prior preference. The MTRL method in Yang et al. (2019) trained and saved models with different fixed prior preferences, and then chooses the best model according to the testing requirement. However, the performance of these approaches highly depends on the selection of the fixed preference, and can also suffer from the conflict among the gradient of different objective functions such that some tasks with larger gradients dominates the update direction at the sacrifice of significant performance degeneration on the less-fortune tasks with smaller gradients. Therefore, it is highly important to find an update direction that aims to find a more balanced solution for all tasks.

There have been a large body of studies on finding a conflict-avoidant (CA) direction to mitigate the gradient conflict among tasks in the context of supervised multi-task learning (MTL). For example, multiple-gradient descent algorithm (MGDA) based methods Chen et al. (2023); Cheng et al. (2023) dynamically updated the weights of tasks such that the deriving direction optimizes all objective functions jointly instead of focusing only on tasks with dominant gradients. The similar idea was then incorporated into various follow-up methods such as CAGrad, PCGrad, Nash-MTL and SDMGrad Yu et al. (2020a); Liu et al. (2021); Navon et al. (2022); Xiao et al. (2023). Although these methods have been also implemented in the MTRL setting, none of them provide a finite-time performance guarantee. Then, an open question arises as:

*Can we develop a dynamic weighting MTRL algorithm, which not only mitigates the gradient conflict among tasks, but also achieves a solid finite-time convergence guarantee?*

However, addressing this question is not easy, primarily due to the difficulty in conducting sample complexity analysis for dynamic weighting MTRL algorithms. This challenge arises from the presence of non-vanishing errors, including optimization errors (e.g., induced by actor-critic) and function approximation error, in gradient estimation within MTRL. However, existing theoretical analysis in the supervised MTL requires the gradient to be either unbiased Xiao et al. (2023); Chen et al. (2023) or diminishing with iteration number Fernando et al. (2022). As a result, the analyses applicable to the supervised setting cannot be directly employed in the MTRL setting, emphasizing the necessity for novel developments in this context. Our specific contributions are summarized as follows.

## 1.1 CONTRIBUTIONS

In this paper, we provide an affirmative answer to the aforementioned question by proposing a novel Multi-Task Actor-Critic (MTAC) algorithm, and further developing the first-known sample complexity analysis for dynamic weighting MTRL.

**Conflict-avoidant Multi-task actor-critic algorithm.** Our proposed MTAC contains three major components: the critic update, the task weight update, and the actor update. First, the critic update is to evaluate policies and then compute the policy gradients for all tasks. Second, we provide two options for updating the task weights. The first option aims to update the task weights such that the weighted direction is close to the CA direction (which is defined as the direction that maximizes the minimum value improvement among tasks). This option enhances the capability of our MTAC to mitigate the gradient conflict among tasks, but at the cost of a slower convergence rate. As a complement, we further provide the second option, which cannot ensure a small CA distance (i.e., the distance to the CA direction as elaborated in Definition 3.1), but allows for a much faster convergence rate. Third, by combining the policy gradients and task weights in the first and second steps, the actor then performs an update on the policy parameter.

**Sample complexity analysis and CA distance guarantee.** We provide a comprehensive sample complexity analysis for the proposed MTAC algorithm under two options for updating task weights, which we refer to as MTAC-CA and MTAC-FC (i.e., MTAC with fast convergence). For MTAC-CA, our analysis shows that it requires $\mathcal{O}(\epsilon^{-5})$ samples per task to attain an $\epsilon + \epsilon_{\mathrm{app}}$-accurate Pareto stationary point (see definition in Definition 3.2), while guaranteeing a small $\epsilon + \sqrt{\epsilon_{\mathrm{app}}}$-level CA distance, where $\epsilon_{\mathrm{app}}$ corresponds to the inherent function approximation error and can be arbitrary small when using a suitable feature function. The analysis for MTAC-FC shows that it can improve the sample complexity of MTAC-FC from $\mathcal{O}(\epsilon^{-5})$ to $\mathcal{O}(\epsilon^{-3})$, but with a constant $\mathcal{O}(1)$-level CA distance. Note that this trade-off between the sampling complexity and CA distance is consistent with the observation in the supervised setting Chen et al. (2023).

Our primary technical contribution lies in the approximation of the CA direction. Instead of directly bounding the gap between the weighted policy gradient $\hat{d}$ and the CA direction $d^*$ as in the supervised setting, which is challenging due to the gradient estimation bias, we construct a surrogate direction $d_s$ that equals to the expectation of $\hat{d}$ to decompose this gap into two distances as $\|d_s - \hat{d}\|$ and $\|d_s - d^*\|$, where the former one can be bounded similarly to the supervised case due to the unbiased estimation, and the latter can be bounded using the critic optimization error and function approximation error together (see Appendix C.1 for more details). This type of analysis may be of independent interest to the theoretical studies for both MTL and MTRL.

**Supportive experiments.** We conduct experiments on the MTRL benchmark MT10 Yu et al. (2020b) and demonstrate that the proposed MTAC-CA algorithm can achieve better performance than existing MTRL algorithms with fixed preference.

## 2 RELATED WORKS

**MTRL.** Existing MTRL algorithms can be mainly categorized into two groups: single-policy MTRL and multi-policy MTRL Vamplew et al. (2011); Liu et al. (2014). Single-policy methods generally aim to find the optimal policy with given *preference* among tasks, and are often sample efficient and easy to implement Yang et al. (2019). However, they may suffer from the issue of gradient conflict among tasks. Multi-policy methods tend to learn a set of policies to approximate the Pareto front. One commonly-used approach is to run a single-policy method for multiple times, each time with a different preference. For example, Zhou et al. (2020) proposed a model-based envelop value iteration (EVI) to explore the Pareto front with a given set of preferences. However, most MTRL works focus on the empirical performance of their methods Iqbal & Sha (2019); Zhang et al. (2021b); Christianos et al. (2022). In this paper, we propose a novel dynamic weighting MTRL method and further provide a sample complexity analysis for it.

**Actor-critic sample complexity analysis.** The sample complexity analysis of the vanilla actor-critic algorithm with linear function approximation have been widely studied Qiu et al. (2021); Kumar et al. (2023); Xu et al. (2020); Barakat et al. (2022); Olshevsky & Gharesifard (2022). These works focus on the single-task RL problem. Some recent works Nian et al. (2020); Reymond et al. (2023); Zhang et al. (2021a) studied multi-task actor-critic algorithms but mainly on their empirical performance. The theoretical analysis of multi-task actor-critic algorithms still remains open.

**Gradient manipulation based MTL and theory.** A variety of MGDA-based methods have been proposed to solve MTL problems because of their simplicity and effectiveness. One of their primal goals is to mitigate the gradient conflict among tasks. For example, PCGrad Yu et al. (2020a) avoided this conflict by projecting the gradient of each task on the norm plane of other tasks. GradDrop Chen et al. (2020) randomly dropped out conflicted gradients. CAGrad Liu et al. (2021) added a constraint on the update direction to be close to the average gradient. Nash-MTL Navon et al. (2022) modeled the MTL problem as a bargain game.

Theoretically, Liu et al. (2014) analyzed the convergence of MGDA for convex objective functions. Fernando et al. (2022) proposed MoCo by estimating the true gradient with a tracking variable, and analyzed its convergence in both the convex and nonconvex settings. Chen et al. (2023) provided a theoretical characterization on the trade-off among optimization, generalization and conflict-avoidance in MTL. Xiao et al. (2023) developed a provable MTL method named SDMGrad based on a double sampling strategy, as well as a preference-oriented regularization. This paper provides the first-known finite-time analysis for such type of methods in the MTRL setting.

## 3 PROBLEM FORMULATION

We first introduce the standard Markov decision processes (MDPs), represented by $\mathcal{M} = (\mathcal{S}, \mathcal{A}, \gamma, P, r)$, where $\mathcal{S}$ and $\mathcal{A}$ are state and action spaces. $\gamma$ is discount factor, $P$ denotes the probability transition kernel, and $r : \mathcal{S} \times \mathcal{A} \rightarrow [0, 1]$ is the reward function. In this paper, we study multi-task reinforcement learning (MTRL) in multi-task MDPs. Each task is associated with a distinct MDP defined as $\mathcal{M}_k = (\mathcal{S}, \mathcal{A}, \gamma, P_k, r_k)$, $k = 0, 1, ..., K - 1$. The tasks have the same state and action spaces but different probability transition kernels and reward functions. The distribution $\xi_0^k$ is the initial state distribution of task $k \in [K]$, where $[K] := \{1, ..., K\}$ and $s_0 \sim \xi_0^k$. Denote by $\mathcal{P} := (\mathcal{S} \times \mathcal{A})^K \rightarrow \Delta(\mathcal{S}^K)$ the joint transition kernel, where $\mathcal{P}(s^{1'}, ..., s^{K'} | (s^1, a^1), ..., (s^K, a^K)) = \Pi_{k \in [K]} P_k(s^{k'} | s^k, a^k)$ and the transition kernels of tasks are independent. A policy $\pi : \mathcal{S} \rightarrow \Delta(\mathcal{A})$ is a mapping from a state to a distribution over the action space, where $\Delta(\mathcal{A})$ is the probability simplex over $\mathcal{A}$. Given a policy $\pi$, the value function of task $k \in [K]$ is defined as:

$$V_\pi^k(s) := \mathbb{E}\left[ \sum_{t=0}^{\infty} \gamma^t r_k(s_t^k, a_t^k) | s_0^k = s, \pi, P_k \right].$$

The action-value function can be defined as:

$$Q_\pi^k(s, a) := \mathbb{E}\left[ \sum_{t=0}^{\infty} \gamma^t (r_k(s_t^k, a_t^k)) | s_0^k = s, a_0^k = a, \pi, P_k \right].$$

Moreover, the visitation distribution induced by the policy $\pi$ of task $k \in [K]$ is defined as $d_\pi^k(s,a) = (1-\gamma)\sum_{t=0}^\infty \gamma^t \mathbb{P}(s_t^k = s, a_t^k = a | s_0^k \sim \xi_0^k, \pi, P^k)$. Denote by $d_\pi \in \Delta((\mathcal{S})^K)$ the joint visitation distribution that $d_\pi(s^1, a^1, ..., s^K, a^K) = (1-\gamma)\sum_{t=0}^\infty \gamma^t \mathbb{P}(s_t^1 = s^1, a_t^1 = a^1, ..., s_t^K = s^K, a_t^K = a^K | s_0^k \sim \xi_0^k(\cdot), \pi, \mathcal{P})$. Then, it can be shown that $d_\pi^k(s,a)$ is the stationary distribution induced by the Markov chain with the transition kernel Konda & Tsitsiklis (2003) $\widetilde{P}(\cdot|s,a) = \gamma P(\cdot|s,a) + (1-\gamma)\xi_0^k(\cdot)$. For a given policy $\pi$, the objective function of task $k \in [K]$ is the expected total discounted reward function: $J^k(\pi) = \mathbb{E}\left[\sum_{t=0}^\infty \gamma^t r_k(s_t^k, a_t^k) | s_0^k \sim \xi_0^k, \pi, P^k\right]$.

In this paper, we parameterize the policy by $\theta \in \Theta$ and get the parameterized policy class $\{\pi_\theta : \theta \in \Theta\}$. Denote by $\psi_\theta(s,a) = \nabla \log \pi_\theta(a|s)$. For convenience, we rewrite $J^k(\theta) = J^k(\pi_\theta)$ and $d_\theta^k = d_{\pi_\theta}^k$. The policy gradient $\nabla J^k(\theta)$ for task $k \in [K]$ is Sutton et al. (1999):

$$\nabla J^k(\theta) = \mathbb{E}_{d_\theta^k}\left[Q_{\pi_\theta}^k(s,a)\psi_\theta(s,a)\right]. \tag{2}$$

In this paper, to address the challenge of large-scale problems, we use linear function approximation to approximate the $Q$ function. Given a policy $\pi_\theta$ parameterized by $\theta \in \mathbb{R}^m$ and feature map $\phi^k : \mathcal{S} \times \mathcal{A} \to \mathbb{R}^m$ for $k \in [K]$, we parameterize the $Q$ function of task $k \in [K]$ by $w^k \in \mathbb{R}^m$, $\widehat{Q}_{\pi_\theta}^k(s,a) := (\phi^k(s,a))^\top w^k$.

**Notations:** The vector $Q(s,a) = \left[Q^k(s,a);\right]_{k\in[K]} \in \mathbb{R}^K$ constitutes the $Q^k(s,a)$ for each task $k \in [K]$ $\left(\text{resp. } V(s) = \left[V^k(s);\right]_{k\in[K]}, J(\pi) = \left[J^k(\pi);\right]_{k\in[K]}\right)$, and the matrix $\boldsymbol{w} = \left[w^k;\right]_{k\in[K]} \in \mathbb{R}^{m \times K}$ constitutes the vector $w^k \in \mathbb{R}^m$ for parameters in each task $k \in [K]$. For a vector $x \in \mathbb{R}^K$, the notation $x \geq 0$ means $x_k \geq 0$ for any $k \in [K]$.

One big issue in MTRL problem is gradient conflict, where gradients for different tasks may vary heavily such that some tasks with larger gradients dominate the update direction at the sacrifice of significant performance degeneration on the less fortune tasks with smaller gradients Yu et al. (2020a). To address this problem, we tend to update the policy in a direction that finds a more balanced solution for all tasks. Specifically, consider a direction $\varrho$, along which we update our policy. We would like to choose $\varrho$ to optimize the value function for every individual task. Toward this goal, we consider the following minimum value improvement among all tasks:

$$\min_{k\in[K]}\left\{\frac{1}{\alpha}\left(J^k(\theta + \alpha\varrho) - J^k(\theta)\right)\right\} \approx \min_{k\in[K]}\left\langle\nabla J^k(\theta), \varrho\right\rangle, \tag{3}$$

where the "$\approx$" holds assuming $\alpha$ is small by applying the first-order Taylor approximation. We would like to find a direction that maximizes the minimum value improvement in 3 among all tasks Désidéri (2012):

$$\max_{\varrho\in\mathbb{R}^m}\min_{k\in[K]}\left\{\frac{1}{\alpha}\left(J^k(\theta+\alpha\varrho) - J^k(\theta)\right)\right\} - \frac{\|\varrho\|^2}{2} \approx \max_{\varrho\in\mathbb{R}^m}\min_{\lambda\in\Lambda}\left\langle\sum_{k=1}^K \lambda^k\nabla J^k(\theta), \varrho\right\rangle - \frac{\|\varrho\|^2}{2}, \tag{4}$$

where $\Lambda$ is the probability simplex over $[K]$. The regularization term $-\frac{1}{2}\|\varrho\|^2$ is introduced here to control the magnitude of the update direction $\varrho$. The solution of the min-max problem in equation 4 can be obtained by solving the following problem Xiao et al. (2023):

$$\varrho^* = (\lambda^*)^\top \nabla J(\theta); s.t. \quad \lambda^* \in \arg\min_{\lambda\in\Lambda}\frac{1}{2}\left\|\lambda^\top \nabla J(\theta)\right\|^2. \tag{5}$$

Once we obtain $\varrho^*$ from equation 5, which is referred to as conflict-avoidant direction, we then update our policy along this direction.

In our MTRL problem, there exist stochastic noise and function approximation error (due to the use of function approximation $\widehat{Q}_{\pi_\theta}^k(s,a) := (\phi^k(s,a))^\top w^k$). Therefore, obtaining the exact solution to equation 5 may not be possible. Denote by $\hat{\varrho}$ the stochastic estimate of $\varrho^*$. We define the following CA distance to measure the divergence between $\hat{\varrho}$ and $\varrho^*$.

**Definition 3.1.** $\|\widehat{\varrho} - \varrho^*\|$ denotes the CA distance at between $\hat{\varrho}$ and $\varrho^*$.

Since conflict-avoidant direction mitigates gradient conflict, the CA distance measures the gap between our stochastic estimate $\widehat{\varrho}$ to the exact solution $\varrho^*$. The larger CA distance is, the further $\widehat{\varrho}$ will be away from $\rho^*$ and more conflict there will be. Thus, it reflects the extent of gradient conflict

of $\widehat{\varrho}$. Our experiments in Table 2 also show that a smaller CA distance yields a more balanced performance among tasks.

Unlike single-task learning RL problems, where any two policies can be easily ordered based on their value functions, in MTRL, one policy could perform better on task $i$, and the other performs better on task $j$. To this end, we need the notion of Pareto stationary point defined as follows.

**Definition 3.2.** If $\mathbb{E}[\min_{\lambda \in \Lambda} \|\lambda^\top \nabla J(\pi)\|^2] \leq \epsilon$, policy $\pi$ is an $\epsilon$-accurate Pareto stationary policy.

In this paper, we will investigate the convergence to a Pareto stationary point and the trade-off between the CA distance and the convergence rate.

## 4 MAIN RESULTS

In this section, we first provide the design of our Multi-Task Actor-Critic (MTAC) algorithm to find a Pareto stationary policy and further present a comprehensive finite sample analysis.

### 4.1 ALGORITHM DESIGN

Our algorithm consists of three major components: (1) critic: policy evaluation via TD(0) to evaluate the current policy (Line 3 to Line 12); (2) stochastic gradient descent (SGD) to update $\lambda$ (Line 13 to Line 14); and (3) actor: policy update along the conflict-avoidant direction (Line 15 to Line 19).

---

**Algorithm 1** Multi-Task Actor-Critic (MTAC)

---

1: **Initialize:** $\theta_0, \boldsymbol{w}_0, \lambda_0, T, N_{\text{actor}}, N_{\text{critic}}, N_{\text{CA}}, N_{\text{FC}}$
2: **for** $t = 0$ **to** $T - 1$ **do**
3:    *Critic Update:*
4:    **for** $k = 1$ **to** $K$ **do**
5:       Sample $(s_0^k, a_0^k) \sim d_t^k$
6:       **for** $j = 0$ **to** $N_{\text{critic}} - 1$ **do**
7:          Observe $s_{j+1}^k \sim \mathbb{P}^k(\cdot|s_j^k, a_j^k), r_j^k$; take action $a_{j+1}^k \sim \pi_{\theta_t}(\cdot|s_{j+1}^k)$
8:          Compute the TD error $\delta_j^k$ according to equation 6
9:          Update $w_{t,j+1}^k = \mathcal{T}_B(w_{t,j}^k + \alpha_{t,j}\delta_j^k \phi^k(s_j^k, a_j^k))$
10:       **end for**
11:    **end for**
12:    Set $\boldsymbol{w}_{t+1} = \boldsymbol{w}_{t, N_{\text{critic}}}$
13:    *Option I: Multi-step update for small CA distance* : $\lambda_{t+1} = \text{CA}(\lambda_t, \pi_{\theta_t}, \boldsymbol{w}_{t+1}, N_{\text{CA}})$
14:    *Option II: Single-step update for fast convergence*: $\lambda_{t+1} = \text{FC}(\lambda_t, \pi_{\theta_t}, \boldsymbol{w}_{t+1}, N_{\text{FC}})$
15:    *Actor Update:*
16:    **for** $k = 1$ **to** $K$ **do**
17:       Independently draw $(s_i^k, a_i^k) \sim d_{\theta_t}^k, i \in [N_{\text{actor}}]$
18:    **end for**
19:    Update policy parameter $\theta_{t+1}$ according to equation 9
20: **end for**

---

**Critic update:** In the critic part, we use TD(0) to evaluate the current policy for all the tasks. Recall that there are $K$ feature functions $\phi^k(\cdot, \cdot), k \in [K]$ for the $K$ tasks. In Line 8 of Algorithm 1, the temporal difference (TD) error of task $k$ at step $j$, $\delta_t^j$, can be calculated based on the critic's estimated $Q$-function of task $k$, $\phi^{k\top} w_{t,j}$ and the reward $r_j^k$ as follows:

$$\delta_j^k = r_j^k + \gamma \langle \phi^k(s_{j+1}^k, a_{j+1}^k), w_{t,j}^k \rangle - \langle \phi^k(s_j^k, a_j^k), w_{t,j}^k \rangle. \tag{6}$$

Then, in Line 9, a TD(0) update is performed, where $\mathcal{T}_B(v) = \arg\min_{\|w\|_2 \leq B} \|v - w\|_2$, $B$ is some positive constant and $\alpha_{t,j}$ is the step size. Such a projection is commonly used in TD algorithms to simplify the analysis, e.g., Qiu et al. (2021); Kumar et al. (2023); Xu et al. (2020); Barakat et al. (2022); Olshevsky & Gharesifard (2022); Zou et al. (2019). After $N$ iterations, we can obtain estimates of $Q$-functions for all tasks.

**Weight $\lambda$ update:** To get the accurate direction of policy gradient in MTRL problems, we solve the problem in equation 5. Recall that there are two targets: small gradient conflict and fast convergence rate. We then provide two different weight update options: multi-step update for small CA distance in Algorithm 2 and single-step update for fast convergence in Algorithm 3.

---

**Algorithm 2** Multi-step update for small CA distance (CA)

1: **Initialize:** $\lambda_t$, $\pi_{\theta_t}$, $\boldsymbol{w}_{t+1}$, $N_{\mathrm{CA}}$; Set $\lambda_{t,0} = \lambda_t$
2: **for** $k = 1$ **to** $K$ **do**
3:     Independently draw $(s_i^k, a_i^k) \sim d_{\theta_t}^k$, $i \in [N_{\mathrm{CA}}]$; $(s_{i'}^k, a_{i'}^k) \sim d_{\theta_t}^k$, $i' \in [N_{\mathrm{CA}}]$
4: **end for**
5: **for** $i = 0$ **to** $N_{\mathrm{CA}} - 1$ **do**
6:     Update $\lambda_{t,i+1}$ according to equation 7
7: **end for**
8: Output $\lambda_{t+1} = \lambda_{t,N_{\mathrm{CA}}}$

---

Firstly, the CA subprocedure independently draws $2N_{\mathrm{CA}}$ state-action pairs following the visitation distribution. The estimated policy gradient of task $k$ by state-action pair $(s_i^k, a_i^k)$

$$\widetilde{\nabla} J_i^k(\theta_t) = \phi^k(s_i^k, a_i^k)^\top w_{t+1}^k \psi_{\theta_t}(s_i^k, a_i^k).$$

Then it uses a projected SGD with a warm start initialization and double-sampling strategy to update the weight $\lambda_t$:

$$\lambda_{t,i+1} = \mathcal{T}_\Lambda \left( \lambda_{t,i} - c_{t,i} \lambda_{t,i}^\top \widetilde{\nabla} J_i(\theta_t) \widetilde{\nabla} J_{i'}(\theta_t)^\top \right), \tag{7}$$

where $c_{t,i}$ is the stepsize, $\widetilde{\nabla} J_i(\theta_t) = \left[ \widetilde{\nabla} J_i^k(\theta_t); \right]_{k \in [K]}$. Weight $\lambda_t$ update $N_{\mathrm{CA}}$ steps in order to obtain a premise estimate of $\lambda_t^* \in \arg\min_{\lambda \in \Lambda} ||\lambda^\top \nabla J(\theta_t)||^2$.

Based on Algorithm 2, we can find a Pareto stationary policy with a small CA distance, but it requires a large sample complexity of $N_{\mathrm{CA}} = \mathcal{O}(\epsilon^{-4})$ as will be shown in Corollary 4.7. However, we sometimes may sacrifice in terms of the CA distance in order for an improved sample complexity. To this end, we also provide an FC subprocedure in Algorithm 3.

---

**Algorithm 3** Single-step update for fast convergence (FC)

1: **Initialize:** $\lambda_t$, $\pi_{\theta_t}$, $\boldsymbol{w}_{t+1}$, $N_{\mathrm{FC}}$
2: **for** $k = 1$ **to** $K$ **do**
3:     Independently draw $(s_i^k, a_i^k) \sim d_{\theta_t}^k$, $i \in [N_{\mathrm{FC}}]$; independently draw $(s_{i'}^k, a_{i'}^k) \sim d_{\theta_t}^k$, $i \in [N_{\mathrm{FC}}]$
4: **end for**
5: Update $\lambda_{t+1}$ according to equation 8 and output $\lambda_{t+1}$

---

In this algorithm, we generate $2N_{\mathrm{FC}}$ samples from the visitation distribution. Alternatively, we only update $\lambda$ once using all the samples in an averaged way:

$$\lambda_{t+1} = \mathcal{T}_\Lambda \left( \lambda_t - c_t \lambda_t^\top \bar{\nabla} J(\theta_t) \bar{\nabla} J(\theta_t)^\top \right), \tag{8}$$

where $\bar{\nabla} J(\theta_t) = \left[ \bar{\nabla} J^k(\theta_t); \right]_{k \in [K]}$ and $\bar{\nabla} J^k(\theta_t) = \frac{1}{N_{\mathrm{FC}}} \sum_{i=0}^{N_{\mathrm{FC}}-1} \phi^k(s_i^k, a_i^k)^\top w_{t+1}^k \psi_{\theta_t}(s_i^k, a_i^k)$ (resp. $\bar{\nabla} J'(\theta_t)$).

As will be shown in Corollary 4.9, to guarantee convergence of the algorithm to a Pareto stationary point, only $N_{\mathrm{FC}} = \mathcal{O}(\epsilon^{-2})$ samples are needed, which is much less than the CA subprocedure. But this is at the price of an increased CA distance.

**Actor update:** For the actor, the policy $\pi_{\theta_t}$ is updated along the conflict-avoidant direction. Given the current estimate of $\lambda_t$, $\theta_t$ and $\omega_t$, the conflict-avoidant direction is a linear combination of policy gradients of all tasks.

In Line 17 of Algorithm 1, $N$ state-action pair $(s_l^k, a_l^k)$, $l = 0, ..., N_{\mathrm{actor}} - 1$, are drawn from the visitation distribution $d_t^k$. Then the policy gradient for task $k$ is estimated as follows:

$$\widetilde{\nabla} J^k(\theta_t) = \frac{1}{N_{\mathrm{actor}}} \sum_{l=0}^{N_{\mathrm{actor}}-1} \phi^k(s_l^k, a_l^k)^\top w_{t+1}^k \psi_{\theta_t}(s^k, a^k).$$

Next, combined with the weight $\lambda_{t+1}$ from Algorithm 2 or Algorithm 3, the policy update direction can be obtained and the policy can be updated by the following rule:

$$\theta_{t+1} = \theta_t + \beta_t \lambda_t^\top \widetilde{\nabla} J(\theta_t). \tag{9}$$

For technical convenience, we assume samples from the visitation distribution induced by the transition kernel and the current policy can be obtained. In practice, the visitation distribution can be simulated by resetting the MDP to the initial state distribution at each time step with probability $1 - \gamma$ Konda & Tsitsiklis (2003), however, this only incur an additional logarithmic factor in the sample complexity.

## 4.2 Theoretical analysis

We first introduce some standard assumptions and then present the finite-sample analysis of our proposed algorithms.

### 4.2.1 Assumptions and definitions

**Assumption 4.1** (Smoothness). let $\pi_\theta(a|s)$ be a policy parameterized by $\theta$. There exist constants $C_\phi = \max\{C_{\phi,1}, C_{\phi,2}\}$ and $C_{\phi,1}, C_{\phi,2}, C_\pi, L_\phi > 0$ and such that

1) $||\nabla \log \pi_\theta(a|s)||_2 \leq C_{\phi,1} \leq C_\phi$;      2) $||\phi^k(s^k, a^k)||_2 \leq C_{\phi,2} \leq C_\phi$ for any $k \in [K]$;

3) $||\pi_\theta(a|s) - \pi_{\theta'}(a|s)||_2 \leq C_\pi ||\theta - \theta'||_2$;      4) $||\log \pi_\theta(a|s) - \log \pi_{\theta'}(a|s)||_2 \leq L_\phi ||\theta - \theta'||_2$.

These assumptions impose the smoothness and boundedness conditions on the policy and feature function, respectively. These assumptions have been widely adopted in the analysis of RL Qiu et al. (2021); Kumar et al. (2023); Xu et al. (2020); Barakat et al. (2022); Olshevsky & Gharesifard (2022), and can be satisfied for many policy classes such as softmax policy class and neural network policy class.

**Assumption 4.2** (Uniform Ergodicity). Consider the MDP with policy $\pi_\theta$ and transition kernel $P^k$, there exist constants $m > 0$, and $\rho \in (0, 1)$ such that

$$\sup_{s \in \mathcal{S}, a \in \mathcal{A}} \left\| \mathbb{P}(s_t, a_t | s_0 = s, \pi_\theta, P^k) - d_{\pi_\theta}^k(\cdot, \cdot) \right\|_{\mathcal{TV}} \leq m\rho^t,$$

where $\|\cdot\|_{\mathcal{TV}}$ denotes the total variation distance between two distributions. This ergodicity assumption has been widely used in theoretical RL to prove the convergence of TD algorithms Qiu et al. (2021); Kumar et al. (2023); Xu et al. (2020); Barakat et al. (2022); Olshevsky & Gharesifard (2022).

Furthermore, we assume that the $m$ feature functions of task $k$, $\phi_i^k, i \in [m], k \in [K]$ are linearly independent. To introduce the function approximation error, we define the matrix $A_{\pi_\theta}^k$ and vector $b_{\pi_\theta}^k$ as follows:

$$A_{\pi_\theta}^k = \mathbb{E}_{d_\theta^k} \left[ \phi(s^k, a^k) \left( \gamma \phi(s^{k'}, a^{k'}) - \phi(s^k, a^k) \right)^\top \right]; \quad b_{\pi_\theta}^k = \mathbb{E}_{d_\theta^k} \left[ \phi(s^k, a^k) R(s^k, a^k) \right]. \quad (10)$$

Denote by $w_\theta^{*k}$ the TD limiting point satisfies:

$$A_{\pi_\theta}^k w_\theta^{*k} + b_{\pi_\theta}^k = \mathbf{0}. \quad (11)$$

**Assumption 4.3** (Problem Solvability). For any $\theta \in \Theta$ and task $k \in [K]$, the matrix $A_{\pi_\theta}^k$ is negative definite and has the maximum eigenvalue of $-\lambda_A$.

Assumption 4.3 is to guarantee solvability of Equation (11) and is widely applied in the literature Wu et al. (2020); Zou et al. (2019); Xu et al. (2020). Then, we define the function approximation error due to the use of linear function approximation in policy evaluation.

**Definition 4.4** (Function Approximation Error). The approximation error of linear function approximation is defined as

$$\epsilon_{\text{app}} = \max_\theta \max_k \sqrt{\mathbb{E}_{d_\theta^k} \left[ \left( \phi^k(s, a)^\top w_\theta^{*k} - Q_{\pi_\theta}^k(s, a) \right)^2 \right]}.$$

We note that the error $\epsilon_{\text{app}}$ is zero if the tabular setting with finite state and action spaces is considered, and can be arbitrarily small with designed feature functions for large/continuous state spaces.

#### 4.2.2 THEORETICAL ANALYSIS FOR MTAC-CA

We first provide an upper-bound on the CA distance for our proposed method.

**Proposition 4.5.** *Suppose Assumptions 4.1 and 4.2 are satisfied. We choose $c_{t,i} = \frac{c}{\sqrt{i}}$, where $c > 0$ is a constant and $i$ is the number of iterations for updating $\lambda_{t,i}$. Then, the CA distance is bounded as:*

$$\|\lambda_{t,N_{CA}}^\top \widehat{\nabla} J_{\boldsymbol{w}_{t+1}}(\theta_t) - (\lambda_t^*)^\top \nabla J(\theta_t)\| \leq \mathcal{O}\Big(\frac{1}{\sqrt[4]{N_{CA}}} + \frac{1}{\sqrt{N_{critic}}} + \sqrt{\epsilon_{app}}\Big),$$

*where $\widehat{\nabla} J_{w_{t+1}}^k(\theta_t) = \mathbb{E}_{d_{\theta_t}^k}[\phi^k(s,a)^\top w_{t+1}^k \psi_{\theta_t}(s,a)], \widehat{\nabla} J_{\boldsymbol{w}_{t+1}}(\theta_t) = \left[\widehat{\nabla} J_{w_{t+1}^k}^k(\theta_t);\right]_{k\in[K]}.$*

Proposition 4.5 shows that the CA distance decreases with the numbers $N_{\text{CA}}$ and $N_{\text{critic}}$ of iterations on $\lambda$'s update. Based on this important characterization, we obtain the convergence result for MTAC-CA.

**Theorem 4.6.** *Suppose Assumptions 4.1 and 4.2 are satisfied. We choose $\beta_t = \beta \leq \frac{1}{L_J}$ as a constant and $\alpha_{t,j} = \frac{1}{2\lambda_A(j+1)}$, $c_{t,i} = \frac{c}{\sqrt{i}}$, where $c > 0$ is a constant. Then, we have*

$$\frac{1}{T}\sum_{t=0}^{T-1}\mathbb{E}[\|(\lambda_t^*)^\top \nabla J(\theta_t)\|^2] = \mathcal{O}\Big(\frac{1}{\beta T} + \epsilon_{app} + \frac{\beta}{N_{actor}} + \frac{1}{\sqrt{N_{critic}}} + \frac{1}{\sqrt[4]{N_{CA}}}\Big).$$

Here $L_J$ is the Lipschitz constant of $\nabla J^k(\theta)$, which can be found in Appendix A. We then characterize the sample complexity and CA distance for the proposed MTAC-CA method in the following corollary.

**Corollary 4.7.** *Under the same setting as in Theorem 4.6, choosing $\beta = \mathcal{O}(1)$, $T = \mathcal{O}(\epsilon^{-1})$, $N_{actor} = \mathcal{O}(\epsilon^{-1})$, $N_{critic} = \mathcal{O}(\epsilon^{-2})$ and $N_{CA} = \mathcal{O}(\epsilon^{-4})$, MTAC-CA finds an $\epsilon + \epsilon_{app}$-accurate Pareto stationary policy while ensuring an $\mathcal{O}(\epsilon + \sqrt{\epsilon_{app}})$ CA distance. Each task uses $\mathcal{O}(\epsilon^{-5})$ samples.*

The above corollary shows that our MTAC-CA algorithm achieves a sample complexity of $\mathcal{O}(\epsilon^{-5})$ to find an $(\epsilon + \epsilon_{\text{app}})$-accurate Pareto stationary policy. Note that this result improves the complexity of $\mathcal{O}(\epsilon^{-6})$ of SDMGrad in the supervised setting. This is because our algorithm draw $\mathcal{O}(N_{\text{critic}} + N_{\text{actor}} + N_{\text{FC}})$ samples to estimate the conflict-avoidant direction, which reduces the variance compared with the approach that only uses one sample.

#### 4.2.3 CONVERGENCE ANALYSIS FOR MTAC-FC

If we could sacrifice a bit on the CA distance, we could further improve the sample complexity to $\mathcal{O}(\epsilon^{-3})$ with the choice of the FC subprocedure.

**Theorem 4.8.** *Suppose Assumption 4.1 and Assumption 4.2 are satisfied. We choose $\beta_t = \beta \leq \frac{1}{L_J}$, $c_t = c' \leq \frac{1}{8C_\phi^2 B}$ as constants, and $\alpha_{t,j} = \frac{1}{2\lambda_A(j+1)}$. Then we have*

$$\frac{1}{T}\sum_{t=0}^{T-1}\mathbb{E}[\|(\lambda_t^*)^\top \nabla J(\theta_t)\|^2] = \mathcal{O}\Big(\frac{1}{\beta T} + \frac{1}{c'T} + \epsilon_{app} + \frac{1}{\sqrt{N_{critic}}} + \frac{\beta}{N_{actor}} + \frac{c'}{N_{FC}}\Big).$$

Though we still need $\mathcal{O}(N_{\text{critic}} + N_{\text{actor}} + N_{\text{FC}})$ samples in Algorithm 3, we do not require an as small CA distance, which helps to improve the sample complexity to $\mathcal{O}(\epsilon^{-3})$ as shown in below.

**Corollary 4.9.** *Under the same setting as in Theorem 4.8, choosing $\beta = \mathcal{O}(1)$, $c' = \mathcal{O}(1)$, $T = \mathcal{O}(\epsilon^{-1})$, $N_{critic} = \mathcal{O}(\epsilon^{-2})$, $N_{actor} = \mathcal{O}(\epsilon^{-1})$, and $N_{FC} = \mathcal{O}(\epsilon^{-1})$, we can achieve an $(\epsilon + \epsilon_{app})$-accurate Pareto stationary policy and each task uses $\mathcal{O}(\epsilon^{-3})$ samples.*

The above corollary shows that our MTAC-FC algorithm achieve a sample complexity of $\mathcal{O}(\epsilon^{-3})$ to find an $(\epsilon + \epsilon_{\text{app}})$-accurate Pareto stationary point. In supervised learning, the fast convergence reach $\mathcal{O}(\epsilon^{-2})$ Xiao et al. (2023) sample size to find $\epsilon$-accurate Pareto stationary policy. This is because the estimation of value function needs more samples.

## 5 PROOF SKETCH (MTAC-CA)

Here, we provide a proof sketch for the convergence and CA distance analysis to highlight major challenges and our technical novelties. We first define $\widehat{\lambda}'_t = \arg\min_{\lambda \in \Lambda} \left\| \lambda^\top \widehat{\nabla} J_{\boldsymbol{w}_{t+1}}(\theta_t) \right\|_2^2$. Recall that

$$\widehat{\nabla} J_{w_{t+1}}^k(\theta_t) = \mathbb{E}_{d_{\theta_t}^k}[\phi^k(s,a)^\top w_{t+1}^k \psi_{\theta_t}(s,a)]; \quad \widehat{\nabla} J_{\boldsymbol{w}_{t+1}}(\theta_t) = \left[\widehat{\nabla} J_{w_{t+1}^k}^k(\theta_t); \right]_{k \in [K]}.$$

The first step is to analyze the convergence for the critic updates and shows that $\mathbb{E}[\|w_{t+1}^k - w_t^{*k}\|^2] = \mathcal{O}\left(\frac{1}{N_{\text{critic}}}\right)$. The next step is to bound the square of the CA distance, which is defined as

$$\|\lambda_{t,N_{\text{CA}}}^\top \widehat{\nabla} J_{\boldsymbol{w}_{t+1}}(\theta_t) - (\lambda_t^*)^\top \nabla J(\theta_t)\|^2.$$

Differently from the supervised setting, the estimator $\widehat{\nabla} J_{\boldsymbol{w}_{t+1}}(\theta_t)$ here is biased due to the presence of the function approximation error. Thus, we need to provide new techniques to control this CA distance, as shown in the following 5 steps.

**Step 1 (Error decomposition):** First, by introducing a surrogate direction $(\widehat{\lambda}'_t)^\top \widehat{\nabla} J_{\boldsymbol{w}_{t+1}}(\theta_t)$ and using the optimality condition that

$$\langle \lambda_{t,N_{\text{CA}}}^\top \nabla J(\theta_t), (\lambda_t^*)^\top \nabla J(\theta_t) \rangle \geq \|(\lambda_t^*)^\top \nabla J(\theta_t)\|^2,$$

the CA distance can be decomposed into three error terms as follows:

$$\|\lambda_{t,N_{\text{CA}}}^\top \widehat{\nabla} J_{\boldsymbol{w}_{t+1}}(\theta_t) - (\lambda_t^*)^\top \nabla J(\theta_t)\|^2 \leq \|\lambda_{t,N_{\text{CA}}}^\top \widehat{\nabla} J_{\boldsymbol{w}_{t+1}}(\theta_t)\|^2 - \|(\widehat{\lambda}'_t)^\top \widehat{\nabla} J_{\boldsymbol{w}_{t+1}}(\theta_t)\|^2$$
$$+ \|(\widehat{\lambda}'_t)^\top \widehat{\nabla} J_{\boldsymbol{w}_{t+1}}(\theta_t)\|^2 - \|(\lambda_t^*)^\top \nabla J(\theta_t)\|^2 - 2\langle \lambda_{t,N_{\text{CA}}}^\top (\widehat{\nabla} J_{\boldsymbol{w}_{t+1}}(\theta_t) - \nabla J(\theta_t)), (\lambda_t^*)^\top \nabla J(\theta_t) \rangle. \tag{12}$$

**Step 2 (Gap between $\lambda_{t,N_{\text{CA}}}$ and $\widehat{\lambda}'_t$):** We bound the error between the direction applied in Algorithm 1 $\|\lambda_{t,N_{\text{CA}}}^\top \widehat{\nabla} J_{\boldsymbol{w}_{t+1}}(\theta_t)\|^2$ and the surrogate direction $\|(\widehat{\lambda}'_t)^\top \widehat{\nabla} J_{\boldsymbol{w}_{t+1}}(\theta_t)\|^2$ (the first line second and third terms in equation 12). Apply the convergence results of SGD, and we can show that this error is of the order $\mathcal{O}(\frac{1}{\sqrt{N_{\text{CA}}}})$.

**Step 3 (Gap between $\widehat{\lambda}'_t$ and $\lambda_t^*$):** In this step, we bound the surrogate direction $\|(\widehat{\lambda}'_t)^\top \widehat{\nabla} J_{\boldsymbol{w}_{t+1}}(\theta_t)\|$ and CA-direction $\|(\lambda_t^*)^\top \nabla J(\theta_t)\|$ (the second line first and second terms in equation 12), which are solutions to minimization problems. The term can be decomposed into the critic error and the function approximation error, and its order is $\mathcal{O}(\frac{1}{N_{\text{critic}}} + \epsilon_{\text{app}})$. This is the technique we use to deal with the gradient bias in MTRL problem.

**Step 4 (Bound on the rest terms):** The rest terms in equation 12 can be easily bounded by the function approximation error and the critic error.

**Step 5:** Combining steps 1-4, we conclude the proof for the CA distance.

Then to show the convergence, we characterize the upper bound of $\left\|(\lambda_t^*)^\top \nabla J(\theta_t)\right\|^2$, which is decomposed into bounds for the CA distance

$$\left\| \lambda_{t,N_{\text{CA}}}^\top \widehat{\nabla} J_{\boldsymbol{w}_{t+1}}(\theta_t) - (\lambda_t^*)^\top \nabla J(\theta_t) \right\|^2,$$

and the surrogate direction $\|\lambda_{t,N_{\text{CA}}}^\top \widehat{\nabla} J_{\boldsymbol{w}_{t+1}}(\theta_t)\|^2$. Those bounds can be derived using the Lipschitz property of the objective function. This completes the proof.

## 6 EXPERIMENTS

We conduct experiments on the MT10 benchmark which includes 10 robotic manipulation tasks from the MetaWorld environment Yu et al. (2020b). The benchmark enables simulated robots to learn a policy that generalizes to a wide range of daily tasks and environments. We adopt soft Actor-Critic (SAC) Haarnoja et al. (2018) as the underlying training algorithm. We compare our algorithms with the single-task learning (STL) with one SAC for each task, Multi-task learning SAC (MTL SAC)

Table 1: Results on MT10 benchmark. Average over 10 random seeds. The success rate and training time per episode are reported.

| Method | Success Rate (mean ± stderr) | Time (Sec.) |
|---|---|---|
| STL | $0.90 \pm 0.03$ | |
| MTL SAC | $0.49 \pm 0.07$ | **3.5** |
| MTL SAC + TE | $0.54 \pm 0.05$ | 4.1 |
| MH SAC | $0.61 \pm 0.04$ | 4.6 |
| Soft Modularization | $0.73 \pm 0.04$ | 7.1 |
| PCGrad | $0.72 \pm 0.02$ | 11.6 |
| MoCo | $0.75 \pm 0.05$ | 11.5 |
| MTAC-CA | $\mathbf{0.81} \pm 0.09$ | 8.3 |
| MTAC-FC | $0.76 \pm 0.11$ | 6.7 |

with a shared model Yu et al. (2020b), Multi-headed SAC (MH SAC) with a shared backbone and task-specific heads Yu et al. (2020b), Multi-task learning SAC with a shared model and task encoder (MTL SAC + TE) Yu et al. (2020b), Soft Modularization Yang et al. (2020) employing a routing network to form task-specific policies. Following the experiment setup in Yu et al. (2020b), we train 2 million steps with a batch size of 1280 and repeat each experiment 10 times over different random seeds. The performance is evaluated once every 10,000 steps and the best average test success rate over the entire training course and average training time (in seconds) per episode is reported. All our experiments are conducted on RTX A6000.

The results are presented in Table 1. Evidently, our proposed MTAC-CA which enjoys the benefit of dynamic weighting outperforms the existing MTRL algorithms with fixed preferences by a large margin. Our algorithm also achieves a better performance than Soft Modularization, which utilizes different policies across tasks. It is demonstrated that the algorithms with fixed preferences are less time-consuming but exhibit poorer performance than Soft Modularization and our algorithms. The results validate that the MTAC-FC is time-efficient with a similar success rate to Soft Modularization.

Table 2: Results of each task on MT10 benchmark. Rate denotes the average success rate over 10 random seeds, and R$i$ ($i = 0, \cdots, 9$) denotes the success rate on each task $i$.

| Steps | Rate | R0 | R1 | R2 | R3 | R4 | R5 | R6 | R7 | R8 | R9 | $\Delta m\% \downarrow$ |
|---|---|---|---|---|---|---|---|---|---|---|---|---|
| 0 | 0.75 | 1.0 | 1.0 | 0.3 | 1.0 | 0.5 | 1.0 | 1.0 | 0.5 | 0.6 | 0.6 | |
| 5 | 0.77 | 1.0 | 0.9 | 0.6 | 1.0 | 0.8 | 1.0 | 1.0 | 0.3 | 0.5 | 0.6 | -9.33 |
| 10 | 0.81 | 1.0 | 0.8 | 0.5 | 1.0 | 0.8 | 1.0 | 1.0 | 0.5 | 0.8 | 0.7 | -15.67 |

As mentioned in Section 4, the CA distance decreases as the number of updates of weight $\lambda$ increases. We adopt 0 steps of update as the baseline and compare it to updating 5 steps and 10 steps. To represent the overall performance of a particular method $m$, we consider using the metric $\Delta m\%$, which is defined as the average per-task performance drop against baseline $b$: $\Delta m\% = \frac{1}{K} \sum_{k=1}^{K} (-1)^{\delta_k} (M_{m,k} - M_{b,k})/M_{b,k} \times 100$, where $M_k$ refers to the $k$-th performance measurement, $M_{b,k}$ represents the result of metric $M_k$ of baseline $b$, $M_{m,k}$ represents the result of metric $M_k$ of method $m$, and $\delta_k = 1$ if a larger value is desired by metric $M_k$. Therefore, a lower value of $\Delta m\%$ indicates that the overall performance is better. Table 2 demonstrates that a smaller CA distance yields more balanced performance.

## 7 CONCLUSION

In this paper, we propose two novel conflict-avoidant multi-task actor-critic algorithms named MTAC-CA and MTAC-FC. We provide a comprehensive convergence rate and sample complexity analysis for both algorithms, and demonstrate the tradeoff between a small CA distance and improved sample complexity. Experiments validate our theoretical results. It is anticipated that our theoretical contribution and the proposed algorithms can be applied to broader MTRL setups.

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

# A    NOTATIONS AND LEMMAS

In this section, we first introduce notations and necessary lemmas in order to help readers understand.

Firstly, we define and recall the notations mas are frequently applied throughout the proof.

We recall that $s^k \in \mathbb{R}^m$ (resp. $a^k$) is the state(action) of task $k$. The bold symbol $\boldsymbol{s} := [s^k;]_{k \in [K]}$ (resp. $\boldsymbol{a} := [a^k;]_{k \in [K]}$). We recall that $\phi^k(s^k, a^k)$ is the feature vector of task $k$ given the state $s^k$ and action $a^k$. The $\phi(\boldsymbol{s}, \boldsymbol{a}) = [\phi^k(s^k, a^k)]_{k \in [K]}$ (resp. $\psi(\boldsymbol{s}, \boldsymbol{a}) = [\psi(s^k, a^k)]_{k \in [K]}$) is the feature vector compose the feature vector of all tasks.

For convenience, denote by $\phi(\boldsymbol{s}, \boldsymbol{a})^\top \boldsymbol{w} = [\phi^k(s^k, a^k)^\top w^k;]_{k \in [K]}$ and $\boldsymbol{\zeta}(\boldsymbol{s}, \boldsymbol{a}, \theta, w) = \langle \phi(\boldsymbol{s}, \boldsymbol{a})^\top \boldsymbol{w}, \psi_\theta(\boldsymbol{s}, \boldsymbol{a}) \rangle = [(\phi^k(s^k, a^k)^\top w^k) \psi_{\theta_t}(s^k, a^k);]_{k \in [K]}$ to help understand.

Next, we introduce necessary lemmas which are widely applied throughout the proof.

**Proposition A.1** (Lipschitz property Xu et al. (2020)). *Under Assumption 4.2 and 4.1, given $\theta, \theta' \in \mathcal{B}$, for any task $k \in [K]$, the objective function satisfies that:*

$$\left\| \nabla J^k(\theta) - \nabla J^k(\theta') \right\|_2 \le L_J \left\| \theta - \theta' \right\|_2,$$

*where $L_J = \frac{1}{(1-\gamma)^2}(4L_\pi C_\phi + L_\phi)$, $L_\pi = \frac{C_\pi}{2}\left(1 + \lceil \log_\rho m \rceil + (1-\rho)^{-1}\right)$.*

Next, we introduce a lemma which is widely used throughout the proof.

**Lemma A.2.** *Suppose there are two functions $f(\cdot)$, $g(\cdot)$ and $x_1^* = \arg\min f(x)$, $x_2^* = \arg\min g(x)$, we have the following inequalities,*

$$|f(x_1^*) - g(x_2^*)| \le \max(|f(x_1^*) - g(x_1^*)|, |f(x_2^*) - g(x_2^*)|).$$

**Lemma A.3.** *For any weight vector $\lambda \in \Lambda$*

$$\sqrt{\mathbb{E}_{d_\theta}\left[\left(\lambda^\top \langle \phi(\boldsymbol{s}, \boldsymbol{a}), \boldsymbol{w}_\theta^* \rangle - \lambda^\top Q_{\pi_\theta}(\boldsymbol{s}, \boldsymbol{a})\right)^2\right]} \le \epsilon_{app}.$$

**Lemma A.4** (MDPs Variance Bound). *Suppose Assumption 4.2 are satisfied, given the policy $\pi_{\theta_t}$ and parameter $\boldsymbol{w}_{t+1}$, sampling $(\boldsymbol{s}_i, \boldsymbol{a}_i) \sim d_{\theta_t}$ i.i.d., $i = 0, 1, ..., N-1$, we can get that*

$$\left| \mathbb{E}\left[ \left\| \frac{1}{N} \sum_{i=0}^{N-1} \lambda_t^\top \boldsymbol{\zeta}(\boldsymbol{s}_i, \boldsymbol{a}_i, \boldsymbol{w}_{t+1}, \theta_t) \right\|_2^2 - \left\| \mathbb{E}_{d_{\theta_t}}\left[ \lambda_t^\top \boldsymbol{\zeta}(\boldsymbol{s}, \boldsymbol{a}, \boldsymbol{w}_{t+1}, \theta_t) \right] \right\|_2^2 \right] \right| \le \frac{2 C_\phi^4 B^2}{N}.$$

Due to the linear function approximation error, the estimation of policy gradients is biased. Based on the biased gradient, the direction of MTRL is biased as well. To bound the bias gap, we define three functions and optimal direction as follows:

$$H_\theta(\lambda) = \|\lambda^\top \mathbb{E}_{d_\theta}[\langle Q_{\pi_\theta}(\boldsymbol{s}, \boldsymbol{a}), \nabla \log \pi_\theta(\boldsymbol{s}, \boldsymbol{a}) \rangle]\|_2$$

$$\lambda_\theta^* = \arg\min_\lambda (H_\theta(\lambda))^2$$

$$\widehat{H}_\theta(\lambda) = \|\lambda^\top \mathbb{E}_{d_\theta}[\langle \phi(\boldsymbol{s}, \boldsymbol{a})^\top \boldsymbol{w}_\theta^*, \nabla \log \pi_\theta(\boldsymbol{s}, \boldsymbol{a}) \rangle]\|_2$$

$$\widehat{\lambda}_\theta^* = \arg\min_\lambda \widehat{H}_\theta^2(\lambda)$$

$$\widehat{H}_\theta'(\lambda) = \|\lambda^\top \mathbb{E}_{d_\theta}[\langle \phi(\boldsymbol{s}, \boldsymbol{a})^\top \boldsymbol{w}_{\theta, N}, \nabla \log \pi_\theta(\boldsymbol{s}, \boldsymbol{a}) \rangle]\|_2$$

$$\widehat{\lambda}_\theta' = \arg\min_\lambda (\widehat{H}_\theta'(\lambda))^2. \tag{13}$$

Here, the first function $H_\theta(\lambda)$ is the unbiased direction loss function and the direction $\lambda_\theta^*$ is the unbiased direction deduced by the unbiased policy gradients. The second function is from the biased estimated direction loss function, where $w_\theta^* = [w_\theta^{*k};]_{k \in [K]}$. The direction $\widehat{\lambda}_\theta^*$ is the biased direction due to approximation error of linear function class. The third function is the direction loss function according to the update rule in Algorithm 1, where $w_{\theta, N}$ is the output after $N$-step Critic update iterations. The direction $\widehat{\lambda}_\theta'$ is the limiting point of equation 7.

For convenience, we rewrite $H_{\theta_t}(\lambda) = H_t(\lambda)$ (resp. $\widehat{H}_{\theta_t}(\lambda) = \widehat{H}_t(\lambda)$, $\widehat{H}_{\theta_t}'(\lambda) = \widehat{H}_t'(\lambda)$) and $\lambda_{\theta_t}^* = \lambda_t^*$ (resp. $\widehat{\lambda}_{\theta_t}^* = \widehat{\lambda}_t^*$ and $\widehat{\lambda}_{\theta_t}' = \widehat{\lambda}_t'$) throughout the following proof.

# B  CRITIC PART: APPROXIMATING THE TD FIXED POINT

In this section, we first provide the convergence analysis of the critic part.

**Lemma B.1** (Approximating TD fixed point). *Suppose Assumption 4.1 and Assumption 4.2 are satisfied, for any task $k \in [K]$, we have*

$$\mathbb{E}[\|w_{t+1}^k - w_t^{*k}\|_2^2] \leq \frac{4B^2}{N_{critic} + 1} + \frac{U_\delta^2 C_\phi^2 \log N_{critic}}{4\lambda_A^2 (N_{critic} + 1)},$$

*where $w_{t+1}^k = w_{t,N}^k$ and $U_\delta = 1 + (1 + \gamma)C_\phi B$.*

*Proof.* The analysis of this term follows from Bhandari et al. (2018). Firstly, we do decomposition of the error term $\|w_{t,j+1}^k - w_t^{*k}\|_2^2$:

$$\|w_{t,j+1}^k - w_t^{*k}\|_2^2 = \|\mathcal{T}_B(w_{t,j}^k + \alpha_{t,j}\delta_j^k \phi^k(s_j^k, a_j^k)) - w_t^{*k}\|_2^2$$

$$\overset{(i)}{\leq} \|w_{t,j}^k + \alpha_{t,j}\delta_j^k \phi^k(s_j^k, a_j^k) - w_t^{*k}\|_2^2$$

$$= \|w_{t,j}^k - w_t^{*k}\|_2^2 + \alpha_{t,j}^2 \|\delta_j^k \phi^k(s_j^k, a_j^k)\|_2^2 + 2\alpha_{t,j}\langle w_{t,j}^k - w_t^{*k}, \delta_j^k \phi^k(s_j^k, a_j^k)\rangle, \tag{14}$$

where $(i)$ follows from the fact that $\|\mathcal{T}_B(x) - y\|_2^2 \leq \|x - y\|_2^2$ when $B$ is a convex set.

We define $\delta^k(s^k, a^k, w, \theta) = R^k(s^k, a^k) + \gamma(\phi^k(s^{k'}, a^{k'}))^\top w - (\phi^k(s^k, a^k))^\top w$. According to the definition of $w_t^{*k}$ in Equation (10) and Equation (11), $w_t^{*k}$ satisfies the following equation:

$$\mathbb{E}_{d_{\theta_t}^k}[\phi^k(s^k, a^k)(R^k(s^k, a^k) + \gamma(\phi^k(s^{k'}, a^{k'}))^\top w_t^{*k} - (\phi^k(s^k, a^k))^\top w_t^{*k})] = 0. \tag{15}$$

We can further get that

$$\mathbb{E}_{d_{\theta_t}^k}[\phi^k(s^k, a^k)\delta^k(s^k, a^k, w_t^*, \theta_t)] = 0.$$

Then for the last term of Equation (14), we take the expectation of it

$$\mathbb{E}[\langle w_{t,j}^k - w_t^{*k}, \delta_j^k \phi^k(s_j^k, a_j^k)\rangle]$$

$$= \mathbb{E}[\langle w_{t,j}^k - w_t^{*k}, \delta_j^k \phi^k(s_j^k, a_j^k) - \mathbb{E}_{d_{\theta_t}^k}[\phi^k(s^k, a^k)\delta^k(s^k, a^k, w_t^*, \theta_t)]\rangle]$$

$$= \mathbb{E}[\langle w_{t,j}^k - w_t^{*k}, \delta_j^k \phi^k(s_j^k, a_j^k) - \mathbb{E}_{d_{\theta_t}^k}[\phi^k(s^k, a^k)\delta^k(s^k, a^k, w_t, \theta_t)]\rangle]$$

$$\quad + \mathbb{E}[\langle w_{t,j}^k - w_t^{*k}, \mathbb{E}_{d_{\theta_t}^k}[\phi^k(s^k, a^k)\delta^k(s^k, a^k, w_t, \theta_t)] - \mathbb{E}_{d_{\theta_t}^k}[\phi^k(s^k, a^k)\delta^k(s^k, a^k, w_t^*, \theta_t)]\rangle]$$

$$\overset{(i)}{\leq} \mathbb{E}[\langle w_{t,j}^k - w_t^{*k}, \mathbb{E}_{d_{\theta_t}^k}[\phi^k(s^k, a^k)\delta^k(s^k, a^k, w_t, \theta_t)] - \mathbb{E}_{d_{\theta_t}^k}[\phi^k(s^k, a^k)\delta^k(s^k, a^k, w_t^*, \theta_t)]\rangle]$$

$$\overset{(ii)}{\leq} -\lambda_A \mathbb{E}[\|w_{t,j}^k - w_t^{*k}\|_2^2],$$

where $(i)$ follows from

$$\mathbb{E}[\langle w_{t,j}^k - w_t^{*k}, \delta_j^k \phi^k(s_j^k, a_j^k) - \mathbb{E}_{d_{\theta_t}^k}[\phi^k(s^k, a^k)\delta^k(s^k, a^k, w_t, \theta_t)]\rangle] = 0,$$

and $(ii)$ follows from that

$$\langle w_{t,j}^k - w_t^{*k}, \mathbb{E}_{d_{\theta_t}^k}[\phi^k(s^k, a^k)\delta^k(s^k, a^k, w_t, \theta_t)] - \mathbb{E}_{d_{\theta_t}^k}[\phi^k(s^k, a^k)\delta^k(s^k, a^k, w_t^*, \theta_t)]\rangle$$

$$= \langle w_{t,j}^k - w_t^{*k}, \mathbb{E}_{d_{\theta_t}^k}[\phi^k(s^k, a^k)\left(\mathbb{E}_{d_{\theta_t}^k}\left[\gamma\phi^k(s^{k'}, a^{k'}) - \phi^k(s^k, a^k)\right]\right)(w_{t,j}^k - w_t^{*k})]\rangle$$

$$= \left(w_{t,j}^k - w_t^{*k}\right)^\top A_t^k \left(w_{t,j}^k - w_t^{*k}\right)$$

$$\overset{(i)}{\leq} -\lambda_A \left\|w_{t,j}^k - w_t^{*k}\right\|_2^2,$$

where we rewrite $A_t^k = A_{\pi_{\theta_t}}^k$ for convenience and $(i)$ follows from Assumption 4.3. Then combining Equation (16) into Equation (14), we can get that

$$\mathbb{E}\left[\left\|w_{t,j+1}^k - w_t^{*k}\right\|_2^2\right] \leq (1 - 2\alpha_{t,j}\lambda_A)\mathbb{E}\left[\left\|w_{t,j}^k - w_t^*\right\|_2^2\right] + \alpha_{t,j}^2 U_\delta^2 C_\phi^2.$$

By setting the learning rate $\alpha_{t,j} = \frac{1}{2\lambda_A(j+1)}$, we can obtain

$$\mathbb{E}\left[\left\|w_{t,j+1}^k - w_t^{*k}\right\|_2^2\right] \leq \frac{j}{j+1}\mathbb{E}\left[\left\|w_{t,j}^k - w_t^*\right\|_2^2\right] + U_\delta^2 C_\phi^2 \frac{1}{4\lambda_A^2}\frac{1}{(j+1)^2}.$$

Then by rearranging the above inequality, we have

$$\mathbb{E}\left[\left\|w_{t+1}^k - w_t^{*k}\right\|_2^2\right] \leq \frac{1}{N_{\text{critic}}+1}\mathbb{E}\left[\left\|w_{t,0}^k - w_t^{*k}\right\|_2^2\right] + \frac{U_\delta^2 C_\phi^2}{4\lambda_A^2(N_{\text{critic}}+1)}\sum_{j=1}^{N_{\text{critic}}+1}\frac{1}{j+1}$$

$$\leq \frac{4B^2}{N_{\text{critic}}+1} + \frac{U_\delta^2 C_\phi^2 \log N_{\text{critic}}}{4\lambda_A^2(N_{\text{critic}}+1)}.$$

The proof is complete. $\qquad\qquad\qquad\qquad\qquad\qquad\qquad\qquad\qquad\qquad\qquad\qquad\qquad\square$

## C  CONVERGENCE ANALYSIS FOR MTAC-CA AND CA DISTANCE ANALYSIS

In this section, we take both CA distance and convergence into consideration with the choice of MTAC-CA.

**Lemma C.1.** *Suppose Assumption 4.1 and Assumption 4.2 are satisfied, we have*

$$\mathbb{E}\left[\left|\widehat{H}_t(\widehat{\lambda}_t^*) - \widehat{H}_t'(\widehat{\lambda}_t')\right|\right] = C_\phi^2\sqrt{\frac{4B^2}{N_{critic}+1} + \frac{U_\delta^2 C_\phi^2 \log N_{critic}}{4\lambda_A^2(N_{critic}+1)}}. \qquad (16)$$

*Proof.* According to Lemma A.2, we can get that

$$\left|\widehat{H}_t(\widehat{\lambda}_t^*) - \widehat{H}_t'(\widehat{\lambda}_t')\right| \leq \max\left\{\left|\widehat{H}_t(\widehat{\lambda}_t^*) - \widehat{H}_t'(\widehat{\lambda}_t^*)\right|, \left|\widehat{H}_t(\widehat{\lambda}_t') - \widehat{H}_t'(\widehat{\lambda}_t')\right|\right\}. \qquad (17)$$

According to the notations in Equation (13), the first term in Equation (17) can be bounded as:

$$\left|\widehat{H}_t(\widehat{\lambda}_t^*) - \widehat{H}_t'(\widehat{\lambda}_t^*)\right|$$

$$= \left|\left\|(\widehat{\lambda}_t^*)^\top \mathbb{E}_{d_{\theta_t}}\left[\boldsymbol{\zeta}(\boldsymbol{s}, \boldsymbol{a}, \boldsymbol{w}_t^*, \theta_t)\right]\right\|_2 - \left\|(\widehat{\lambda}_t^*)^\top \mathbb{E}_{d_{\theta_t}}\left[\boldsymbol{\zeta}(\boldsymbol{s}, \boldsymbol{a}, \boldsymbol{w}_{t+1}, \theta_t)\right]\right\|_2\right|$$

$$\leq \left\|(\widehat{\lambda}_t^*)^\top \mathbb{E}_{d_{\theta_t}}\left[\boldsymbol{\zeta}(\boldsymbol{s}, \boldsymbol{a}, \boldsymbol{w}_t^* - \boldsymbol{w}_{t+1}, \theta_t)\right]\right\|_2$$

$$\leq \max_{k\in[K]}\left\|\mathbb{E}_{d_{\theta_t}}\left[\phi^k(s^k, a^k)^\top(w^{*k} - w_{t+1}^k)\psi_{\theta_t}(s^k, a^k)\right]\right\|_2$$

$$\overset{(i)}{\leq} \max_{k\in[K]}\left\{\|\phi^k(s^k, a^k)\|_2\|w_t^{*k} - w_{t+1}^k\|_2\|\psi_{\theta_t}(s^k, a^k)\|_2\right\}$$

$$\overset{(ii)}{\leq} \max_{k\in[K]} C_\phi^2\|w_t^{*k} - w_{t+1}^k\|_2$$

$$= C_\phi^2 \max_{k\in[K]}\|w_t^{*k} - w_{t+1}^k\|_2,$$

where $(i)$ follows from Cauchy-Schwartz inequality and $(ii)$ follows from Assumption 4.1.Thus, taking expectation on both sides, we can obtain,

$$\mathbb{E}\left[\left|\widehat{H}_t(\widehat{\lambda}_t^*) - \widehat{H}_t'(\widehat{\lambda}_t^*)\right|\right] \leq C_\phi^2 \max_{k\in[K]}\mathbb{E}\left[\left\|w_t^{*k} - w_{t+1}^k\right\|_2\right] \leq C_\phi^2 \max_{k\in[K]}\sqrt{\mathbb{E}[\|w_t^{*k} - w_{t+1}^k\|_2^2]}.$$

Similarly, we can get that

$$\mathbb{E}\left[\left|\widehat{H}_t(\widehat{\lambda}_t') - \widehat{H}_t'(\widehat{\lambda}_t')\right|\right] \leq C_\phi^2 \max_{k\in[K]}\sqrt{\mathbb{E}[\|w_t^{*k} - w_{t+1}^k\|_2^2]}.$$

Then, combined with Lemma B.1, we can derive

$$\mathbb{E}\left[\left|\widehat{H}_t(\widehat{\lambda}_t^*) - \widehat{H}_t'(\widehat{\lambda}_t')\right|\right] \leq C_\phi^2\sqrt{\frac{4B^2}{N_{\text{critic}}+1} + \frac{U_\delta^2 C_\phi^2 \log N_{\text{critic}}}{4\lambda_A^2(N_{\text{critic}}+1)}}.$$

The proof is complete. $\qquad\qquad\qquad\qquad\qquad\qquad\qquad\qquad\qquad\qquad\qquad\qquad\qquad\square$

**Lemma C.2.** *Suppose Assumption 4.1 and Assumption 4.2 are satisfied, we have*

$$\mathbb{E}\left[\left|H_t(\lambda_t^*) - \widehat{H_t}(\widehat{\lambda}_t^*)\right|\right] \le 2C_\phi \epsilon_{app},$$

*where $\epsilon_{app}$ is defined in Definition 4.4.*

*Proof.* First we apply Lemma A.2,

$$\left|H_t(\lambda_t^*) - \widehat{H_t}(\widehat{\lambda}_t^*)\right| \le \max\left\{\left|H_t(\lambda_t^*) - \widehat{H_t}(\lambda_t^*)\right|, \left|H_t(\widehat{\lambda}_t^*) - \widehat{H_t}(\widehat{\lambda}_t^*)\right|\right\}.$$

Then for the first term in the above equation,

$$\left|H_t(\lambda_t^*) - \widehat{H_t}(\lambda_t^*)\right|$$

$$= \left|\left\|(\lambda_t^*)^\top \nabla J(\theta_t)\right\|_2 - \left\|(\lambda_t^*)^\top \widehat{\nabla} J_{\boldsymbol{w}_t^*}(\theta_t)\right\|_2\right|$$

$$\le \left\|(\lambda_t^*)^\top \mathbb{E}_{d_{\theta_t}}[\langle Q_{\pi_{\theta_t}}(\boldsymbol{s},\boldsymbol{a}) - \phi^\top(\boldsymbol{s},\boldsymbol{a})\boldsymbol{w}_t^*, \psi_{\theta_t}(\boldsymbol{s},\boldsymbol{a})\rangle]\right\|_2$$

$$\le \max_k\left\{\left\|\mathbb{E}_{d_{\theta_t}}\left[(Q_{\pi_{\theta_t}}^k(s^k,a^k) - \phi^k(s^k,a^k)^\top w_t^{*k})\psi_{\pi_{\theta_t}}(s^k,a^k)\right]\right\|_2\right\}$$

$$\overset{(i)}{\le} C_\phi \max_k\left\{\left\|\mathbb{E}_{d_{\theta_t}}[Q_{\pi_{\theta_t}}^k(s^k,a^k) - \langle\phi^k(s^k,a^k), w_t^{*k}\rangle]\right\|_2\right\}$$

$$\le C_\phi \max_k \sqrt{\mathbb{E}_{d_{\theta_t}}\left[\|Q_{\pi_{\theta_t}}^k(s^k,a^k) - \langle\phi^k(s^k,a^k), w_t^{*k}\rangle\|_2^2\right]}$$

$$\overset{(ii)}{\le} C_\phi \epsilon_{app},$$

where $(i)$ follows from Assumption 4.1 and $(ii)$ follows from Definition 4.4. Then for the term $\left|H_t(\widehat{\lambda}_t^*) - \widehat{H_t}(\widehat{\lambda}_t^*)\right|$, we can follow similar steps and the following inequality can be derived

$$\left|H_t(\widehat{\lambda}_t^*) - \widehat{H_t}(\widehat{\lambda}_t^*)\right| \le C_\phi \epsilon_{app}.$$

Therefore, we can obtain

$$\mathbb{E}\left[\left|H_t(\lambda_t^*) - \widehat{H_t}(\widehat{\lambda}_t^*)\right|\right] \le 2C_\phi \epsilon_{app}.$$

The proof is complete. □

## C.1 PROOF OF PROPOSITION 4.5

**CA distance.** Now we show the upper bound for the distance to CA direction. Recall that we define the CA distance as $\left\|\lambda_{t,N_{CA}}^\top \widehat{\nabla} J_{\boldsymbol{w}_{t+1}}(\theta_t) - (\lambda_t^*)^\top \nabla J(\theta_t)\right\|_2^2$,

$$\|\lambda_{t,N_{CA}}^\top \widehat{\nabla} J_{\boldsymbol{w}_{t+1}}(\theta_t) - (\lambda_t^*)^\top \nabla J(\theta_t)\|^2$$

$$= \|\mathbb{E}_{d_{\theta_t}}[\lambda_{t,N_{CA}}^\top \langle\phi^\top(\boldsymbol{s},\boldsymbol{a})\boldsymbol{w}_{t+1}, \psi_{\theta_t}(\boldsymbol{s},\boldsymbol{a})\rangle] - \mathbb{E}_{d_{\theta_t}}[(\lambda_t^*)^\top Q^{\pi_t}(\boldsymbol{s},\boldsymbol{a})\psi_{\theta_t}(\boldsymbol{s},\boldsymbol{a})]\|_2^2$$

$$= \|\mathbb{E}_{d_{\theta_t}}[\lambda_{t,N_{CA}}^\top \langle\phi^\top(\boldsymbol{s},\boldsymbol{a})\boldsymbol{w}_{t+1}, \psi_{\theta_t}(\boldsymbol{s},\boldsymbol{a})\rangle]\|_2^2 + \|\mathbb{E}_{d_{\theta_t}}[(\lambda_t^*)^\top Q^{\pi_t}(\boldsymbol{s},\boldsymbol{a})\psi_{\theta_t}(\boldsymbol{s},\boldsymbol{a})]\|_2^2$$

$$\quad - 2\langle\mathbb{E}_{d_{\theta_t}}[\lambda_{t,N_{CA}}^\top \langle\phi^\top(\boldsymbol{s},\boldsymbol{a})\boldsymbol{w}_{t+1}, \psi_{\theta_t}(\boldsymbol{s},\boldsymbol{a})\rangle], \mathbb{E}_{d_{\theta_t}}[(\lambda_t^*)^\top Q^{\pi_t}(\boldsymbol{s},\boldsymbol{a})\psi_{\theta_t}(\boldsymbol{s},\boldsymbol{a})]\rangle$$

$$= \|\mathbb{E}_{d_{\theta_t}}[\lambda_{t,N_{CA}}^\top \langle\phi^\top(\boldsymbol{s},\boldsymbol{a})\boldsymbol{w}_{t+1}, \psi_{\theta_t}(\boldsymbol{s},\boldsymbol{a})\rangle]\|_2^2 + \|\mathbb{E}_{d_{\theta_t}}[(\lambda_t^*)^\top Q^{\pi_t}(\boldsymbol{s},\boldsymbol{a})\psi_{\theta_t}(\boldsymbol{s},\boldsymbol{a})]\|_2^2$$

$$\quad - 2\langle\mathbb{E}_{d_{\theta_t}}[\lambda_{t,N_{CA}}^\top Q^{\pi_t}(\boldsymbol{s},\boldsymbol{a})\psi_{\theta_t}(\boldsymbol{s},\boldsymbol{a})], \mathbb{E}_{d_{\theta_t}}[(\lambda_t^*)^\top Q^{\pi_t}(\boldsymbol{s},\boldsymbol{a})\psi_{\theta_t}(\boldsymbol{s},\boldsymbol{a})]\rangle$$

$$\quad - 2\langle\mathbb{E}_{d_{\theta_t}}[\lambda_{t,N_{CA}}^\top(\langle\phi(\boldsymbol{s},\boldsymbol{a}), \boldsymbol{w}_{t+1}\rangle - Q^{\pi_t}(\boldsymbol{s},\boldsymbol{a}))\psi_{\theta_t}(\boldsymbol{s},\boldsymbol{a})], \mathbb{E}_{d_{\theta_t}}[(\lambda_t^*)^\top Q^{\pi_t}(\boldsymbol{s},\boldsymbol{a})\psi_{\theta_t}(\boldsymbol{s},\boldsymbol{a})]\rangle$$

$$\overset{(i)}{\le} \|\mathbb{E}_{d_{\theta_t}}[\lambda_{t,N_{CA}}^\top \langle\phi^\top(\boldsymbol{s},\boldsymbol{a})\boldsymbol{w}_{t+1}, \psi_{\theta_t}(\boldsymbol{s},\boldsymbol{a})\rangle]\|_2^2 - \|\mathbb{E}_{d_{\theta_t}}[(\lambda_t^*)^\top Q^{\pi_t}(\boldsymbol{s},\boldsymbol{a})\psi_{\theta_t}(\boldsymbol{s},\boldsymbol{a})]\|_2^2$$

$$\quad - 2\langle\mathbb{E}_{d_{\theta_t}}[\lambda_{t,N_{CA}}^\top(\langle\phi(\boldsymbol{s},\boldsymbol{a}), \boldsymbol{w}_{t+1}\rangle - Q^{\pi_t}(\boldsymbol{s},\boldsymbol{a}))\psi_{\theta_t}(\boldsymbol{s},\boldsymbol{a})], \mathbb{E}_{d_{\theta_t}}[(\lambda_t^*)^\top Q^{\pi_t}(\boldsymbol{s},\boldsymbol{a})\psi_{\theta_t}(\boldsymbol{s},\boldsymbol{a})]\rangle$$

$$\le \underbrace{\left\|\mathbb{E}_{d_{\theta_t}}\left[\lambda_{t,N_{CA}}^\top \langle\phi^\top(\boldsymbol{s},\boldsymbol{a})\boldsymbol{w}_{t+1}, \psi_{\theta_t}(\boldsymbol{s},\boldsymbol{a})\rangle\right]\right\|_2^2 - \left\|\mathbb{E}_{d_{\theta_t}}\left[(\widehat{\lambda}_t')^\top \langle\phi^\top(\boldsymbol{s},\boldsymbol{a})\boldsymbol{w}_{t+1}, \psi_{\theta_t}(\boldsymbol{s},\boldsymbol{a})\rangle\right]\right\|_2^2}_{\text{term I}}$$

$$+ \|\mathbb{E}_{d_{\theta_t}}[(\widehat{\lambda}'_t)^\top \langle \phi^\top(\boldsymbol{s}, \boldsymbol{a})\boldsymbol{w}_{t+1}, \psi_{\theta_t}(\boldsymbol{s}, \boldsymbol{a})\rangle]\|_2^2 - \|\mathbb{E}_{d_{\theta_t}}[(\lambda^*_t)^\top Q^{\pi_t}(\boldsymbol{s}, \boldsymbol{a})\psi_{\theta_t}(\boldsymbol{s}, \boldsymbol{a})]\|_2^2$$

$$- 2\langle \mathbb{E}_{d_{\theta_t}}[\lambda^\top_{t, N_{\text{CA}}}((\langle \phi(\boldsymbol{s}, \boldsymbol{a}), \boldsymbol{w}_{t+1}\rangle - Q^{\pi_t}(\boldsymbol{s}, \boldsymbol{a}))\psi_{\theta_t}(\boldsymbol{s}, \boldsymbol{a})], \mathbb{E}_{d_{\theta_t}}[(\lambda^*_t)^\top Q^{\pi_t}(\boldsymbol{s}, \boldsymbol{a})\psi_{\theta_t}(\boldsymbol{s}, \boldsymbol{a})]\rangle,$$

where $(i)$ follows from the optimality condition that

$$\langle \lambda_{t, N_{\text{CA}}}, (Q^{\pi_t}(\boldsymbol{s}, \boldsymbol{a})\psi_{\theta_t}(\boldsymbol{s}, \boldsymbol{a})^\top Q^{\pi_t}(\boldsymbol{s}, \boldsymbol{a})\psi_{\theta_t}(\boldsymbol{s}, \boldsymbol{a})\lambda^*_t\rangle$$

$$\geq \langle \lambda^*_t, (Q^{\pi_t}(\boldsymbol{s}, \boldsymbol{a})\psi_{\theta_t}(\boldsymbol{s}, \boldsymbol{a})^\top Q^{\pi_t}(\boldsymbol{s}, \boldsymbol{a})\psi_{\theta_t}(\boldsymbol{s}, \boldsymbol{a})\lambda^*_t\rangle = \|(\lambda^*_t)^\top Q^{\pi_t}(\boldsymbol{s}, \boldsymbol{a})\psi_{\theta_t}(\boldsymbol{s}, \boldsymbol{a})\|_2^2. \quad (18)$$

Next, we bound the term I as follows:

term I

$$= \left\|\mathbb{E}_{d_{\theta_t}}\left[\lambda^\top_{t, N_{\text{CA}}} \langle \phi^\top(\boldsymbol{s}, \boldsymbol{a})\boldsymbol{w}_{t+1}, \psi_{\theta_t}(\boldsymbol{s}, \boldsymbol{a})\rangle\right]\right\|_2^2 - \left\|\mathbb{E}_{d_{\theta_t}}\left[(\widehat{\lambda}'_t)^\top \langle \phi^\top(\boldsymbol{s}, \boldsymbol{a})\boldsymbol{w}_{t+1}, \psi_{\theta_t}(\boldsymbol{s}, \boldsymbol{a})\rangle\right]\right\|_2^2$$

$$\overset{(i)}{\leq} \left(\frac{2}{c} + 2cC_1\right)\frac{2 + \log N_{\text{CA}}}{\sqrt{N_{\text{CA}}}}, \quad (19)$$

where $(i)$ follows from Theorem 2 Shamir & Zhang (2013) since the gradient estimator is unbiased, $\sup_{\lambda, \lambda'} \|\lambda - \lambda'\| \leq 1$, $\mathbb{E}[\|(\phi(\boldsymbol{s}, \boldsymbol{a})^\top \boldsymbol{w}_{t+1}\psi_{\theta_t}(\boldsymbol{s}, \boldsymbol{a}))^\top(\phi(\boldsymbol{s}, \boldsymbol{a})^\top \boldsymbol{w}_{t+1}\psi_{\theta_t}(\boldsymbol{s}, \boldsymbol{a})\widehat{\lambda}'_t\|] \leq C_\phi^4 B^2 = C_1$, and $c_{t,i} = \frac{c}{\sqrt{i}}$. Then, the last term can be bounded as follows:

$$\|\mathbb{E}_{d_{\theta_t}}[(\widehat{\lambda}'_t)^\top \langle \phi^\top(\boldsymbol{s}, \boldsymbol{a})\boldsymbol{w}_{t+1}, \psi_{\theta_t}(\boldsymbol{s}, \boldsymbol{a})\rangle]\|_2^2 - \|\mathbb{E}_{d_{\theta_t}}[(\lambda^*_t)^\top Q^{\pi_t}(\boldsymbol{s}, \boldsymbol{a})\psi_{\theta_t}(\boldsymbol{s}, \boldsymbol{a})]\|_2^2$$

$$- 2\langle \mathbb{E}_{d_{\theta_t}}[\lambda^\top_{t, N_{\text{CA}}}((\langle \phi(\boldsymbol{s}, \boldsymbol{a}), \boldsymbol{w}_{t+1}\rangle - Q^{\pi_t}(\boldsymbol{s}, \boldsymbol{a}))\psi_{\theta_t}(\boldsymbol{s}, \boldsymbol{a})], \mathbb{E}_{d_{\theta_t}}[(\lambda^*_t)^\top Q^{\pi_t}(\boldsymbol{s}, \boldsymbol{a})\psi_{\theta_t}(\boldsymbol{s}, \boldsymbol{a})]\rangle$$

$$\overset{(i)}{\leq} \left|\|\mathbb{E}_{d_{\theta_t}}[(\widehat{\lambda}'_t)^\top \langle \phi^\top(\boldsymbol{s}, \boldsymbol{a})\boldsymbol{w}_{t+1}, \psi_{\theta_t}(\boldsymbol{s}, \boldsymbol{a})\rangle]\|_2 - \|\mathbb{E}_{d_{\theta_t}}[(\lambda^*_t)^\top Q^{\pi_t}(\boldsymbol{s}, \boldsymbol{a})\psi_{\theta_t}(\boldsymbol{s}, \boldsymbol{a})]\|_2\right|$$

$$\times (\|\mathbb{E}_{d_{\theta_t}}[(\widehat{\lambda}'_t)^\top \langle \phi^\top(\boldsymbol{s}, \boldsymbol{a})\boldsymbol{w}_{t+1}, \psi_{\theta_t}(\boldsymbol{s}, \boldsymbol{a})\rangle]\|_2 + \|\mathbb{E}_{d_{\theta_t}}[(\lambda^*_t)^\top Q^{\pi_t}(\boldsymbol{s}, \boldsymbol{a})\psi_{\theta_t}(\boldsymbol{s}, \boldsymbol{a})]\|_2)$$

$$+ 2\|\mathbb{E}_{d_{\theta_t}}[\lambda^\top_{t, N_{\text{CA}}}((\langle \phi(\boldsymbol{s}, \boldsymbol{a}), \boldsymbol{w}_{t+1}\rangle - Q^{\pi_t}(\boldsymbol{s}, \boldsymbol{a}))\psi_{\theta_t}(\boldsymbol{s}, \boldsymbol{a})]\|_2 \|\mathbb{E}_{d_{\theta_t}}[(\lambda^*_t)^\top Q^{\pi_t}(\boldsymbol{s}, \boldsymbol{a})\psi_{\theta_t}(\boldsymbol{s}, \boldsymbol{a})]\|_2$$

$$\overset{(ii)}{\leq} C_\phi^4 B^2 |\widehat{H}'_t(\widehat{\lambda}'_t) - H_t(\lambda^*_t)| + \frac{2C_\phi^2}{1 - \gamma}\epsilon_{\text{app}}$$

$$\overset{(iii)}{\leq} C_\phi^6 B^2 \sqrt{\frac{4B^2}{N_{\text{critic}} + 1} + \frac{U_\delta^2 C_\phi^2 \log N_{\text{critic}}}{4\lambda_A^2 (N_{\text{critic}} + 1)}} + \frac{2C_\phi^2}{1 - \gamma}\epsilon_{\text{app}},$$

where $(i)$ follows from the inequality that $\|A\|_2^2 - \|B\|_2^2 \leq \|\|A\|_2 - \|B\|_2|(\|A\|_2 + \|B\|_2)$ and Cauchy-Schwartz inequality. $(ii)$ follows from the definition in Equation (13). $(iii)$ follows from Lemma C.1. Then we apply Lemma C.2, we can derive

$$\|\mathbb{E}_{d_{\theta_t}}[\lambda^\top_{t, N_{\text{CA}}} \langle \phi^\top(\boldsymbol{s}, \boldsymbol{a})\boldsymbol{w}_{t+1}, \psi_{\theta_t}(\boldsymbol{s}, \boldsymbol{a})\rangle] - \mathbb{E}_{d_{\theta_t}}[(\lambda^*_t)^\top Q^{\pi_t}(\boldsymbol{s}, \boldsymbol{a})\psi_{\theta_t}(\boldsymbol{s}, \boldsymbol{a})]\|$$

$$= \mathcal{O}\left(\frac{1}{\sqrt[4]{N_{\text{CA}}}} + \frac{1}{\sqrt{N_{\text{critic}}}} + \sqrt{\epsilon_{\text{app}}}\right).$$

The proof is complete.

**Theorem C.3** (Restatement of Theorem 4.6). *Suppose Assumption 4.1 and Assumption 4.2 are satisfied. We choose $\beta_t = \beta \leq \frac{1}{L_J}$ as a constant and $c_{t,i} = \frac{c}{\sqrt{i}}$ where $i$ is the iteration number for updating $\lambda_{t,i}$, and we have*

$$\frac{1}{T}\sum_{t=0}^{T-1}\mathbb{E}[\|(\lambda^*_t)^\top \nabla J(\theta_t)\|_2^2] = \mathcal{O}\left(\frac{1}{\beta T} + \epsilon_{app} + \frac{\beta}{N_{actor}} + \frac{1}{\sqrt{N_{critic}}} + \frac{1}{\sqrt[4]{N_{CA}}}\right).$$

*Proof.* We first define a fixed simplex $\bar{\lambda} = [\bar{\lambda}_1, \bar{\lambda}_2, ..., \bar{\lambda}_K]$. According to the Proposition A.1, for each task $k \in [K]$, we have

$$J^k(\theta_t) \leq J^k(\theta_{t+1}) - \langle \nabla J^k(\theta_t), \theta_{t+1} - \theta_t\rangle + \frac{L_J}{2}\|\theta_{t+1} - \theta_t\|_2^2,$$

where $k \in [K]$. Then by multiplying $\bar{\lambda}_k$ on both sides and summing over $k$, we can obtain,

$$\bar{\lambda}^\top J(\theta_t) \leq \bar{\lambda}^\top J(\theta_{t+1}) - \langle \bar{\lambda}^\top \nabla J(\theta_t), \theta_{t+1} - \theta_t\rangle + \frac{L_J}{2}\|\theta_{t+1} - \theta_t\|_2^2, \quad (20)$$

then recalling from Algorithm 1, we have the update rule

$$\theta_{t+1} = \theta_t + \beta_t \frac{1}{N_{\text{actor}}} \sum_{l=0}^{N_{\text{actor}}-1} \lambda_{t+1}^\top \langle \phi(\boldsymbol{s}_l, \boldsymbol{a}_l)^\top \boldsymbol{w}_{t+1}, \psi_{\theta_t}(\boldsymbol{s}_l, \boldsymbol{a}_l) \rangle$$

$$= \theta_t + \beta_t \frac{1}{N_{\text{actor}}} \sum_{l=0}^{N_{\text{actor}}-1} \lambda_{t+1}^\top \boldsymbol{\zeta}(\boldsymbol{s}_l, \boldsymbol{a}_l, \theta_t, \boldsymbol{w}_{t+1}).$$

Thus for the third term, we have

$$\mathbb{E}[\|\theta_{t+1} - \theta_t\|_2^2] = \mathbb{E}[\|\theta_{t+1} - \theta_t\|_2^2 - \beta_t^2 \|\lambda_{t+1}^\top \mathbb{E}[\boldsymbol{\zeta}(\boldsymbol{s}, \boldsymbol{a}, \theta_t, \boldsymbol{w}_{t+1})]\|_2^2]$$
$$+ \beta_t^2 \|\lambda_{t+1}^\top \mathbb{E}[\boldsymbol{\zeta}(\boldsymbol{s}, \boldsymbol{a}, \theta_t, \boldsymbol{w}_{t+1})]\|_2^2$$
$$\leq \beta_t^2 \mathbb{E}\left[\left\|\frac{1}{N_{\text{actor}}} \sum_{l=0}^{N_{\text{actor}}-1} \boldsymbol{\zeta}(\boldsymbol{s}_l, \boldsymbol{a}_l, \theta_t, \boldsymbol{w}_{t+1})\right\|_2^2 - \|\lambda_{t+1}^\top \mathbb{E}[\boldsymbol{\zeta}(\boldsymbol{s}, \boldsymbol{a}, \theta_t, \boldsymbol{w}_{t+1})]\|_2^2\right]$$
$$+ \beta_t^2 \|\mathbb{E}[\boldsymbol{\zeta}(\boldsymbol{s}, \boldsymbol{a}, \theta_t, \boldsymbol{w}_{t+1})]\|_2^2$$
$$\overset{(i)}{\leq} \beta_t^2 \frac{2C_\phi^4 B^2}{N_{\text{actor}}} + \beta_t^2 \|\mathbb{E}[\boldsymbol{\zeta}(\boldsymbol{s}, \boldsymbol{a}, \theta_t, \boldsymbol{w}_{t+1})]\|_2^2, \tag{21}$$

where $(i)$ follows from Lemma A.4. Then for the second term in Equation (20), we take the expectation of it,

$$-\mathbb{E}[\langle \bar{\lambda}^\top \nabla J(\theta_t), \theta_{t+1} - \theta_t \rangle]$$
$$= -\mathbb{E}[\langle \bar{\lambda}^\top \mathbb{E}_{d_{\theta_t}}[Q_{\pi_{\theta_t}}(\boldsymbol{s}, \boldsymbol{a})\psi_{\theta_t}(\boldsymbol{s}, \boldsymbol{a})], \theta_{t+1} - \theta_t \rangle]$$
$$= -\mathbb{E}[\langle \mathbb{E}_{d_{\theta_t}}[\bar{\lambda}^\top (Q_{\pi_{\theta_t}}(\boldsymbol{s}, \boldsymbol{a}) - \langle \phi(\boldsymbol{s}, \boldsymbol{a}), \boldsymbol{w}_t^* \rangle)\psi_{\theta_t}(\boldsymbol{s}, \boldsymbol{a})], \theta_{t+1} - \theta_t \rangle]$$
$$\quad -\mathbb{E}[\langle \mathbb{E}_{d_{\theta_t}}[\bar{\lambda}^\top \langle \phi(\boldsymbol{s}, \boldsymbol{a}), \boldsymbol{w}_t^* - \boldsymbol{w}_{t+1} \rangle \psi_{\theta_t}(\boldsymbol{s}, \boldsymbol{a})], \theta_{t+1} - \theta_t \rangle]$$
$$\quad -\mathbb{E}[\langle \mathbb{E}_{d_{\theta_t}}[\bar{\lambda}^\top \langle \phi(\boldsymbol{s}, \boldsymbol{a})^\top \boldsymbol{w}_{t+1}, \psi_{\theta_t}(\boldsymbol{s}, \boldsymbol{a}) \rangle], \theta_{t+1} - \theta_t \rangle]$$
$$\overset{(i)}{\leq} \beta_t C_\phi^3 B |\mathbb{E}_{d_{\theta_t}}[Q_{\pi_{\theta_t}}(\boldsymbol{s}, \boldsymbol{a}) - \langle \phi(\boldsymbol{s}, \boldsymbol{a}), \boldsymbol{w}_t^* \rangle]| + \beta_t C_\phi^4 B \max_{k \in [K]} \mathbb{E}[\|w_t^{*k} - w_{t+1}^k\|_2]$$
$$\quad - \beta_t \mathbb{E}[\langle \bar{\lambda}^\top \mathbb{E}_{d_{\theta_t}}[\langle \phi(\boldsymbol{s}, \boldsymbol{a})^\top \boldsymbol{w}_{t+1}, \psi_{\theta_t}(\boldsymbol{s}, \boldsymbol{a}) \rangle], \mathbb{E}_{d_{\theta_t}}[\lambda_{t+1}^\top \langle \phi(\boldsymbol{s}, \boldsymbol{a})^\top \boldsymbol{w}_{t+1}, \psi_{\theta_t}(\boldsymbol{s}, \boldsymbol{a}) \rangle]\rangle]$$
$$\leq \beta_t C_\phi^3 B \sqrt{\|\mathbb{E}_{d_{\theta_t}}[Q_{\pi_{\theta_t}}(\boldsymbol{s}, \boldsymbol{a}) - \langle \phi(\boldsymbol{s}, \boldsymbol{a}), \boldsymbol{w}_t^* \rangle]\|_2^2} + \beta_t C_\phi^4 B \max_{k \in [K]} \mathbb{E}[\|w_t^{*k} - w_{t+1}^k\|_2]$$
$$\quad - \beta_t \mathbb{E}[\langle \bar{\lambda}^\top \mathbb{E}_{d_{\theta_t}}[\langle \phi(\boldsymbol{s}, \boldsymbol{a})^\top \boldsymbol{w}_{t+1}, \psi_{\theta_t}(\boldsymbol{s}, \boldsymbol{a}) \rangle], \mathbb{E}_{d_{\theta_t}}[(\widehat{\lambda}_t')^\top \langle \phi(\boldsymbol{s}, \boldsymbol{a})^\top \boldsymbol{w}_{t+1}, \psi_{\theta_t}(\boldsymbol{s}, \boldsymbol{a}) \rangle]\rangle]$$
$$\quad + \beta_t \mathbb{E}[\langle \bar{\lambda}^\top \mathbb{E}_{d_{\theta_t}}[\langle \phi(\boldsymbol{s}, \boldsymbol{a})^\top \boldsymbol{w}_{t+1}, \psi_{\theta_t}(\boldsymbol{s}, \boldsymbol{a}) \rangle], \mathbb{E}_{d_{\theta_t}}[(\widehat{\lambda}_t' - \lambda_{t+1})^\top \langle \phi(\boldsymbol{s}, \boldsymbol{a})^\top \boldsymbol{w}_{t+1}, \psi_{\theta_t}(\boldsymbol{s}, \boldsymbol{a}) \rangle]\rangle]$$
$$\overset{(ii)}{\leq} \beta_t C_\phi^3 B \epsilon_{\text{app}} + \beta_t C_\phi^4 B \sqrt{\frac{4B^2}{N_{\text{critic}}+1} + \frac{U_\delta^2 C_\phi^2 \log N_{\text{critic}}}{4\lambda_A^2(N_{\text{critic}}+1)}} - \beta_t \mathbb{E}[\|\widehat{\lambda}_t' \langle \phi(\boldsymbol{s}, \boldsymbol{a})\boldsymbol{w}_{t+1}\rangle \psi_{\theta_t}(\boldsymbol{s}, \boldsymbol{a})\|_2^2]$$
$$\quad + \beta_t \mathbb{E}[C_\phi^2 B \|(\widehat{\lambda}_t' - \lambda_{t+1})^\top \langle \phi(\boldsymbol{s}, \boldsymbol{a})^\top \boldsymbol{w}_{t+1}, \psi_{\theta_t}(\boldsymbol{s}, \boldsymbol{a}) \rangle\|_2], \tag{22}$$

where $(i)$ follows from Assumption 4.1, $(ii)$ follows from Lemma A.3, Lemma B.1 and optimality condition

$$\mathbb{E}[\langle \bar{\lambda}^\top \mathbb{E}_{d_{\theta_t}}[\langle \phi(\boldsymbol{s}, \boldsymbol{a})^\top \boldsymbol{w}_{t+1}, \psi_{\theta_t}(\boldsymbol{s}, \boldsymbol{a}) \rangle], \mathbb{E}_{d_{\theta_t}}[(\widehat{\lambda}_t')^\top \langle \phi(\boldsymbol{s}, \boldsymbol{a})^\top \boldsymbol{w}_{t+1}, \psi_{\theta_t}(\boldsymbol{s}, \boldsymbol{a}) \rangle]\rangle]$$
$$\geq \mathbb{E}[\|\widehat{\lambda}_t' \langle \phi(\boldsymbol{s}, \boldsymbol{a})^\top \boldsymbol{w}_{t+1}, \psi_{\theta_t}(\boldsymbol{s}, \boldsymbol{a}) \rangle\|_2^2].$$

Again, according to the Theorem 2 in Shamir & Zhang (2013) following the same choice of step size $c_{t,i}$ in Equation (19), we can obtain,

$$\mathbb{E}[\|(\widehat{\lambda}_t' - \lambda_{t+1})^\top \langle \phi(\boldsymbol{s}, \boldsymbol{a})^\top \boldsymbol{w}_{t+1}, \psi_{\theta_t}(\boldsymbol{s}, \boldsymbol{a}) \rangle\|_2^2]$$
$$\leq \mathbb{E}[\|\lambda_{t+1}^\top \mathbb{E}_{d_{\theta_t}}[\langle \phi(\boldsymbol{s}, \boldsymbol{a})^\top \boldsymbol{w}_{t+1}, \psi_{\theta_t}(\boldsymbol{s}, \boldsymbol{a}) \rangle]\|_2^2] - \mathbb{E}[\|(\widehat{\lambda}_t')^\top \mathbb{E}_{d_{\theta_t}}[\langle \phi(\boldsymbol{s}, \boldsymbol{a})^\top \boldsymbol{w}_{t+1}, \psi_{\theta_t}(\boldsymbol{s}, \boldsymbol{a}) \rangle]\|_2^2]$$
$$\leq \left(\frac{2}{c} + 2cC_1\right) \frac{2 + \log N_{\text{CA}}}{\sqrt{N_{\text{CA}}}}.$$

Thus, we can derive

$$-\mathbb{E}[\langle \bar{\lambda}^\top \nabla J(\theta_t), \theta_{t+1} - \theta_t \rangle]$$

$$\leq \beta_t C_\phi^3 B \epsilon_{\text{app}} + \beta_t C_\phi^4 B \sqrt{\frac{4B^2}{N_{\text{critic}}+1} + \frac{U_\delta^2 C_\phi^2 \log N_{\text{critic}}}{4\lambda_A^2(N_{\text{critic}}+1)}} + \beta_t C_\phi^2 B \sqrt{\left(\frac{2}{c} + 2cC_1\right)\frac{2 + \log N_{\text{CA}}}{\sqrt{N_{\text{CA}}}}}$$

$$- \beta_t \mathbb{E}[\|\widehat{\lambda}'_t \langle \phi(s,a)^\top w_{t+1}, \psi_{\theta_t}(s,a)\rangle\|_2^2]. \tag{23}$$

Then combining Equation (23) and Equation (21) into Equation (20), we can obtain that,

$$\mathbb{E}[\bar{\lambda}^\top J(\theta_t)] \leq \mathbb{E}[\bar{\lambda}^\top J(\theta_{t+1})] - \beta_t \mathbb{E}[\|\widehat{\lambda}'_t \mathbb{E}[\langle \phi^\top(s,a)w_{t+1}, \psi_{\theta_t}(s,a)\rangle]\|_2^2]$$

$$+ \frac{L_J \beta_t^2}{2}\mathbb{E}[\|\widehat{\lambda}'_t \langle \phi(s,a)^\top w_{t+1}, \psi_{\theta_t}(s,a)\rangle\|_2^2] + \beta_t^2 \frac{L_J C_\phi^4 B^2}{N_{\text{actor}}} + \beta_t C_\phi^3 B \epsilon_{\text{app}}$$

$$+ \beta_t C_\phi^4 B \sqrt{\frac{4B^2}{N_{\text{critic}}+1} + \frac{U_\delta^2 C_\phi^2 \log N_{\text{critic}}}{4\lambda_A^2(N_{\text{critic}}+1)}} + \beta_t C_\phi^2 B \sqrt{\left(\frac{2}{c} + 2cC_1\right)\frac{2 + \log N_{\text{CA}}}{\sqrt{N_{\text{CA}}}}}. \tag{24}$$

We set $\beta_t = \beta \leq \frac{1}{L_J}$ as a constant. Then, we rearrange and telescope over $t = 0, 1, 2, ..., T-1$,

$$\frac{1}{T}\sum_{t=0}^{T} \mathbb{E}[\|\widehat{\lambda}'_t \mathbb{E}[\langle \phi^\top(s,a)w_{t+1}, \psi_{\theta_t}(s,a)\rangle]\|_2^2] \leq \frac{2}{\beta T}\mathbb{E}[\bar{\lambda}^\top J(\theta_T) - \bar{\lambda}^\top J(\theta_0)] + \beta \frac{2L_J C_\phi^4 B^2}{N_{\text{actor}}}$$

$$+ 2C_\phi^3 B \epsilon_{\text{app}} + 2C_\phi^4 B \sqrt{\frac{4B^2}{N_{\text{critic}}+1} + \frac{U_\delta^2 C_\phi^2 \log N_{\text{critic}}}{4\lambda_A^2(N_{\text{critic}}+1)}} + 2C_\phi^2 B \sqrt{\left(\frac{2}{c} + 2cC_1\right)\frac{2 + \log N_{\text{CA}}}{\sqrt{N_{\text{CA}}}}}. \tag{25}$$

Then we consider our target $\mathbb{E}[\|\lambda_t^* \nabla J(\theta_t)\|_2^2]$, we can derive

$$\mathbb{E}[\|(\lambda_t^*)^\top \nabla J(\theta_t)\|_2^2]$$

$$= \mathbb{E}[\|(\lambda_t^*)^\top \nabla J(\theta_t)\|_2^2] - \mathbb{E}[\|(\widehat{\lambda}_t^*)^\top \mathbb{E}_{d_{\theta_t}}[\langle \phi(s,a)^\top w_t^*, \psi_{\theta_t}(s,a)\rangle]\|_2^2]$$

$$+ \mathbb{E}[\|(\widehat{\lambda}_t^*)^\top \mathbb{E}_{d_{\theta_t}}[\langle \phi(s,a)^\top w_t^*, \psi_{\theta_t}(s,a)\rangle]\|_2^2] - \mathbb{E}[\|(\widehat{\lambda}'_t)^\top \mathbb{E}_{d_{\theta_t}}[\langle \phi(s,a)^\top w_t^*, \psi_{\theta_t}(s,a)\rangle]\|_2^2]$$

$$+ \mathbb{E}[\|(\widehat{\lambda}'_t)^\top \mathbb{E}_{d_{\theta_t}}[\langle \phi(s,a)^\top w_{t+1} \psi_{\theta_t}(s,a)\rangle]\|_2^2]$$

$$\leq 2C_\phi^2 B(|H_t^*(\lambda_t^*) - \widehat{H}_t(\widehat{\lambda}_t^*)| + |\widehat{H}_t(\widehat{\lambda}_t^*) - \widehat{H}'_t(\widehat{\lambda}'_t)|)$$

$$+ \mathbb{E}[\|(\widehat{\lambda}'_t)^\top \mathbb{E}_{d_{\theta_t}}[\langle \phi(s,a)^\top w_{t+1}, \psi_{\theta_t}(s,a)\rangle]\|_2^2]. \tag{26}$$

Then summing over $t = 0, 1, 2, ..., T-1$ of the above inequality, we can get

$$\frac{1}{T}\sum_{t=0}^{T-1} \mathbb{E}[\|(\lambda_t^*)^\top \nabla J(\theta_t)\|_2^2]$$

$$\leq \frac{1}{T}\sum_{t=0}^{T-1} \left(2C_\phi^2 B(|H_t^*(\lambda_t^*) - \widehat{H}_t(\widehat{\lambda}_t^*)| + |\widehat{H}_t(\widehat{\lambda}_t^*) - \widehat{H}'_t(\widehat{\lambda}'_t)|)\right)$$

$$+ \frac{1}{T}\sum_{t=0}^{T-1} \mathbb{E}\left[\|(\widehat{\lambda}'_t)^\top \mathbb{E}_{d_{\theta_t}}[\langle \phi(s,a)^\top w_t^*, \psi_{\theta_t}(s,a)\rangle]\|_2^2\right]$$

$$\overset{(i)}{\leq} 2C_\phi^4 B \sqrt{\frac{4B^2}{N_{\text{critic}}+1} + \frac{U_\delta^2 C_\phi^2 \log N_{\text{critic}}}{4\lambda_A^2(N_{\text{critic}}+1)}} + 2C_\phi^3 B \epsilon_{\text{app}}$$

$$+ \frac{1}{T}\sum_{t=0}^{T-1} \mathbb{E}\left[\|(\widehat{\lambda}'_t)^\top \mathbb{E}_{d_{\theta_t}}[\langle \phi(s,a)^\top w_t^*, \psi_{\theta_t}(s,a)\rangle]\|_2^2\right]$$

$$\overset{(ii)}{\leq} \frac{2}{\beta T}\mathbb{E}[\bar{\lambda}^\top J(\theta_0) - \bar{\lambda}^\top J(\theta_T)] + 2C_\phi^4 B \sqrt{\frac{4B^2}{N_{\text{critic}}+1} + \frac{U_\delta^2 C_\phi^2 \log N_{\text{critic}}}{4\lambda_A^2(N_{\text{critic}}+1)}} + \beta\frac{2L_J C_\phi^4 B^2}{N_{\text{actor}}}$$

$$+ 4C_\phi^3 B\epsilon_{\text{app}} + 2C_\phi^4 B\sqrt{\frac{4B^2}{N_{\text{critic}}+1} + \frac{U_\delta^2 C_\phi^2 \log N_{\text{critic}}}{4\lambda_A^2(N_{\text{critic}}+1)}} + 2C_\phi^2 B\sqrt{\left(\frac{2}{c} + 2cC_1\right)\frac{2+\log N_{\text{CA}}}{\sqrt{N_{\text{CA}}}}},$$

where $(i)$ follows from the Lemmas C.1 and C.2 and $(ii)$ follows from the Equation (26). Lastly, above all, we can derive

$$\frac{1}{T}\sum_{t=0}^{T-1}\mathbb{E}[\|(\lambda_t^*)^\top \nabla J(\theta_t)\|_2^2] = \mathcal{O}\left(\frac{1}{\beta T} + \epsilon_{\text{app}} + \frac{\beta}{N_{\text{actor}}} + \frac{1}{\sqrt{N_{\text{critic}}}} + \frac{1}{\sqrt[4]{N_{\text{CA}}}}\right).$$

The proof is complete. $\qquad\square$

## C.2 PROOF OF COROLLARY 4.7

Since we choose $\beta = \mathcal{O}(1)$, we have

$$\frac{1}{T}\sum_{t=0}^{T-1}\mathbb{E}[\|(\lambda_t^*)^\top \nabla J(\theta_t)\|^2] = \mathcal{O}\left(\frac{1}{\beta T} + \epsilon_{\text{app}} + \frac{\beta}{N_{\text{actor}}} + \frac{1}{\sqrt{N_{\text{critic}}}} + \frac{1}{\sqrt[4]{N_{\text{CA}}}}\right).$$

To achieve an $\epsilon$-accurate Pareto stationary policy, it requires $N_{\text{CA}} = \mathcal{O}(\epsilon^{-4})$, $N_{\text{critic}} = \mathcal{O}(\epsilon^{-2})$, $N_{\text{actor}} = \mathcal{O}(\epsilon^{-1})$, and $T = \mathcal{O}(\epsilon^{-1})$ and each objective requires $\mathcal{O}(\epsilon^{-5})$ samples. Meanwhile, according to the choice of $N_{\text{actor}}$, $N_{\text{critic}}$, $N_{\text{CA}}$, and $T$, CA distance takes the order of $\mathcal{O}(\epsilon + \sqrt{\epsilon_{\text{app}}})$ simultaneously.

## D CONVERGENCE ANALYSIS FOR MTAC-FC

When we do not have requirements on CA distance, we can have a much lower sample complexity. In Algorithm 1, CA subprocedure for $\lambda_t$ update is to reduce the CA distance, which increases the sample complexity. Thus, we will choose Algorithm 3 to make Algorithm 1 more sample-efficient.

### D.1 PROOF OF THEOREM 4.8

**Theorem D.1** (Restatement of Theorem 4.8). *Suppose Assumption 4.1 and Assumption 4.2 are satisfied. We choose $\beta_t = \beta \leq \frac{1}{L_J}$, $c_t = c' \leq \frac{1}{8C_\phi^2 B}$, and $\alpha_{t,j} = \frac{1}{2\lambda_a(j+1)}$ as constant, and we have*

$$\frac{1}{T}\sum_{t=0}^{T-1}\mathbb{E}[\|(\lambda_t^*)^\top \nabla J(\theta_t)\|_2^2] = \mathcal{O}\left(\frac{1}{\beta T} + \frac{1}{c'T} + \epsilon_{app} + \frac{1}{\sqrt{N_{critic}}} + \frac{\beta}{N_{actor}} + \frac{c'}{N_{FC}}\right).$$

*Proof.* According to the descent lemma, we have for any task $k \in [K]$,

$$J^k(\theta_t) \geq J^k(\theta_{t+1}) + \langle \nabla J^k(\theta_t), \theta_{t+1} - \theta_t\rangle - \frac{L_J}{2}\|\theta_{t+1} - \theta_t\|_2^2. \qquad (27)$$

Then we multiply fix weight $\bar{\lambda}^k$ on both sides and sum all inequalities, we can obtain

$$\bar{\lambda}^\top J(\theta_t) \geq \bar{\lambda}^\top J(\theta_{t+1}) + \langle \bar{\lambda}^\top \nabla J(\theta_t), \theta_{t+1} - \theta_t\rangle - \frac{L_J}{2}\|\theta_{t+1} - \theta_t\|_2^2$$

$$= \bar{\lambda}^\top J(\theta_{t+1}) + \beta_t\left\langle \bar{\lambda}^\top \nabla J(\theta_t), \frac{1}{N_{\text{actor}}}\sum_{l=0}^{N_{\text{actor}}-1}\lambda_t^\top \left\langle \phi(s_l, a_l)^\top w_{t+1}, \psi_{\theta_t}(s_l, a_l)\right\rangle\right\rangle$$

$$- \frac{L_J}{2}\beta_t^2\left\|\frac{1}{N_{\text{actor}}}\sum_{l=0}^{N_{\text{actor}}-1}\lambda_t^\top \left\langle \phi(s_l, a_l)^\top w_{t+1}, \psi_{\theta_t}(s_l, a_l)\right\rangle\right\|_2^2.$$

Then following the similar steps in Equation (21), we can get

$$\bar{\lambda}^\top J(\theta_{t+1})$$

$$\geq \bar{\lambda}^\top J(\theta_t) + \beta_t\left\langle \bar{\lambda}^\top \nabla J(\theta_t), \lambda_t^\top \left(\mathbb{E}_{d_{\pi_\theta}}[\langle \phi(s, a)^\top w_t^*, \psi_{\theta_t}(s, a)\rangle] - \nabla J(\theta_t)\right)\right\rangle + \beta_t\left\langle \bar{\lambda}^\top \nabla J(\theta_t),\right.$$

$$\lambda_t^\top \mathbb{E}_{d_{\theta_t}}[\langle \phi(\boldsymbol{s},\boldsymbol{a})^\top \boldsymbol{w}_t^*, \psi_\theta(\boldsymbol{s},\boldsymbol{a})\rangle] - \lambda_t^\top \mathbb{E}_{d_{\theta_t}}[\langle \phi(\boldsymbol{s},\boldsymbol{a})^\top \boldsymbol{w}_{t+1}, \psi_\theta(\boldsymbol{s},\boldsymbol{a})\rangle]\Big\rangle + \beta_t$$

$$\left\langle \bar{\lambda}^\top \nabla J(\theta_t), \frac{1}{N_{\text{actor}}} \sum_{l=0}^{N_{\text{actor}}-1} \lambda_t^\top \langle \phi(\boldsymbol{s}_l,\boldsymbol{a}_l)^\top \boldsymbol{w}_{t+1}, \psi_{\theta_t}(\boldsymbol{s}_l,\boldsymbol{a}_l)\rangle - \lambda_t^\top \mathbb{E}_{d_{\theta_t}}[\langle \phi(\boldsymbol{s},\boldsymbol{a})^\top \boldsymbol{w}_t^*, \psi_\theta(\boldsymbol{s},\boldsymbol{a})\rangle]\right\rangle$$

$$+ \beta_t \langle \bar{\lambda}^\top \nabla J(\theta_t), \lambda_t^\top \nabla J(\theta_t)\rangle - \frac{L_J}{2}\beta_t^2 \left\|\frac{1}{N_{\text{actor}}} \sum_{l=0}^{N_{\text{actor}}-1} \lambda_t^\top \langle \phi(\boldsymbol{s}_l,\boldsymbol{a}_l)^\top \boldsymbol{w}_{t+1}, \psi_{\theta_t}(\boldsymbol{s}_l,\boldsymbol{a}_l)\rangle\right\|_2^2$$

$$\geq \bar{\lambda}^\top J(\theta_{t+1}) + \beta_t \left\langle \bar{\lambda}^\top \nabla J(\theta_t), \lambda_t^\top \left(\mathbb{E}_{d_{\pi_\theta}}[\langle \phi(\boldsymbol{s},\boldsymbol{a})^\top \boldsymbol{w}_t^*, \psi_{\theta_t}(\boldsymbol{s},\boldsymbol{a})\rangle] - \nabla J(\theta_t)\right)\right\rangle + \beta_t \Big\langle \bar{\lambda}^\top \nabla J(\theta_t),$$

$$\frac{1}{N_{\text{actor}}} \sum_{l=0}^{N_{\text{actor}}-1} \lambda_t^\top \langle \phi(\boldsymbol{s}_l,\boldsymbol{a}_l)^\top \boldsymbol{w}_{t+1}, \psi_{\theta_t}(\boldsymbol{s}_l,\boldsymbol{a}_l)\rangle - \lambda_t^\top \mathbb{E}_{d_{\theta_t}}[\langle \phi(\boldsymbol{s},\boldsymbol{a})^\top \boldsymbol{w}_t^*, \psi_\theta(\boldsymbol{s},\boldsymbol{a})\rangle]\Big\rangle$$

$$+ \beta_t \langle \bar{\lambda}^\top \nabla J(\theta_t), \lambda_t^\top \nabla J(\theta_t)\rangle - \beta_t \frac{C_\phi^2}{1-\gamma} \max_k \left\{\left\|w_{t+1}^k - w^{*k}\right\|_2\right\}$$

$$- \frac{L_J}{2}\beta_t^2 \left\|\frac{1}{N_{\text{actor}}} \sum_{l=0}^{N_{\text{actor}}-1} \lambda_t^\top \langle \phi(\boldsymbol{s}_l,\boldsymbol{a}_l)^\top \boldsymbol{w}_{t+1}, \psi_{\theta_t}(\boldsymbol{s}_l,\boldsymbol{a}_l)\rangle\right\|_2^2.$$

Then we take expectations on both sides

$$\mathbb{E}[\bar{\lambda}^\top J(\theta_t)] \overset{(i)}{\geq} \mathbb{E}[\bar{\lambda}^\top J(\theta_{t+1})] + \beta_t \mathbb{E}[\langle \bar{\lambda}^\top \nabla J(\theta_t), \lambda_t^\top \mathbb{E}_{d_{\theta_t}}[\langle \phi(\boldsymbol{s},\boldsymbol{a})^\top \boldsymbol{w}_t^*, \psi_{\theta_t}(\boldsymbol{s},\boldsymbol{a})\rangle] - \lambda_t^\top \nabla J(\theta_t)\rangle]$$

$$+ \beta_t \mathbb{E}[\|\lambda_t^\top \nabla J(\theta_t)\|_2^2] + \beta_t \mathbb{E}[\langle(\bar{\lambda}-\lambda_t)^\top \nabla J(\theta_t), \lambda_t^\top \nabla J(\theta_t)] - \frac{C_\phi^2 \beta_t}{1-\gamma} \max_k \left\{\mathbb{E}\left[\left\|w_{t+1}^k - w^{*k}\right\|_2\right]\right\}$$

$$- \frac{L_J \beta_t^2}{2}\mathbb{E}\left[\left\|\frac{1}{N_{\text{actor}}} \sum_{l=0}^{N_{\text{actor}}-1} \lambda_t \langle \phi(\boldsymbol{s}_l,\boldsymbol{a}_l)^\top \boldsymbol{w}_{t+1}, \psi_{\theta_t}(\boldsymbol{s}_l,\boldsymbol{a}_l)\rangle\right\|_2^2\right]$$

$$\overset{(ii)}{\geq} \mathbb{E}[\bar{\lambda}^\top J(\theta_{t+1})] - \beta_t \underbrace{\mathbb{E}[\langle \bar{\lambda}^\top \nabla J(\theta_t), \lambda_t^\top \nabla J(\theta_t) - \lambda_t^\top \mathbb{E}_{d_{\theta_t}}[\langle \phi(\boldsymbol{s},\boldsymbol{a})^\top \boldsymbol{w}_t^*, \psi_{\theta_t}(\boldsymbol{s},\boldsymbol{a})\rangle]\rangle]}_{\text{term I}}$$

$$- \beta_t \underbrace{\mathbb{E}[\langle(\lambda_t - \bar{\lambda})^\top \nabla J(\theta_t), \lambda_t^\top \nabla J(\theta_t)\rangle]}_{\text{term II}} - \frac{C_\phi^2 \beta_t}{1-\gamma}\left(\sqrt{\frac{4B^2}{N_{\text{critic}}+1} + \frac{U_\delta^2 C_\phi^2 \log N_{\text{critic}}}{4\lambda_A^2(N_{\text{critic}}+1)}}\right)$$

$$+ \beta_t \mathbb{E}[\|\lambda_t^\top \nabla J(\theta_t)\|_2^2] - \beta_t^2 \frac{L_J C_\phi^4 B^2}{N_{\text{actor}}} - \frac{L_J \beta_t^2}{2}\|\mathbb{E}_{d_{\theta_t}}[\lambda_t^\top \langle \phi(\boldsymbol{s},\boldsymbol{a})^\top \boldsymbol{w}_{t+1}, \psi_{\theta_t}(\boldsymbol{s},\boldsymbol{a})\rangle]\|_2^2, \qquad (28)$$

where $(i)$ follows from that

$$\mathbb{E}\Big[\Big\langle \bar{\lambda}^\top \nabla J(\theta_t), \frac{1}{N_{\text{actor}}}$$

$$\sum_{l=0}^{N_{\text{actor}}-1} \lambda_t^\top \langle \phi(\boldsymbol{s}_l,\boldsymbol{a}_l)^\top \boldsymbol{w}_{t+1}, \psi_{\theta_t}(\boldsymbol{s}_l,\boldsymbol{a}_l)\rangle - \lambda_t^\top \mathbb{E}_{d_{\pi_\theta}}[\langle \phi(\boldsymbol{s},\boldsymbol{a})^\top \boldsymbol{w}_{t+1}, \psi_\theta(\boldsymbol{s},\boldsymbol{a})\rangle]\Big\rangle\Big] = 0.$$

$(ii)$ follows from Lemma A.4.

Then, we bound the term I as follows:

$$\text{term I} \leq \max_{k \in [K]}\left\{\mathbb{E}\left[\left\|\nabla J^k(\theta_t)\right\|_2 \mathbb{E}_{d_{\pi_{\theta_t}}^k}\left[\left\|\phi^k(s^k,a^k)^\top w_{t+1}^k - Q_{\pi_{\theta_t}}^k(s^k,a^k)\right\|_2 \left\|\psi_{\theta_t}(s^k,a^k)\right\|_2\right]\right]\right\}$$

$$\overset{(i)}{\leq} \frac{C_\phi^2}{1-\gamma} \max_{k \in [K]}\left\{\sqrt{\mathbb{E}\left[\mathbb{E}_{d_{\pi_{\theta_t}}^k}\left[\left\|\phi^k(s^k,a^k)^\top w_{t+1}^k - Q_{\pi_{\theta_t}}^k(s^k,a^k)\right\|_2^2\right]\right]}\right\}$$

$$\overset{(ii)}{\leq} \frac{C_\phi^2 \epsilon_{\text{app}}}{1-\gamma}, \qquad (29)$$

where $(i)$ follows from that $\left\|\nabla J^k(\theta_t)\right\|_2 = \left\|\mathbb{E}_{d^k_{\pi_{\theta_t}}} \left[Q^k_{\pi_{\theta_t}}(s^k, a^k)\psi_{\theta_t}(s^k, a^k)\right]\right\|_2 \leq C_\phi \frac{1}{1-\gamma}$. $(ii)$ follows from Definition 4.4.

Then, consider the term II, we have

$$
\text{term II} = \mathbb{E}[\langle \lambda_t - \bar{\lambda}, (\nabla J(\theta_t))^\top (\lambda_t^\top \nabla J(\theta_t))\rangle]
$$

$$
= \mathbb{E}\left[\left\langle \lambda_t - \bar{\lambda}, \left(\nabla J(\theta_t) - \frac{1}{N_{\text{FC}}} \sum_{j=0}^{N_{\text{FC}}-1} \langle \phi(s_j, a_j)^\top w_t^*, \psi_{\theta_t}(a_j|s_j)\rangle\right)^\top (\lambda_t^\top \nabla J(\theta_t))\right\rangle\right]
$$

$$
+ \mathbb{E}\left[\left\langle \lambda_t - \bar{\lambda}, \left(\frac{1}{N_{\text{FC}}} \sum_{j=0}^{N_{\text{FC}}-1} \langle \phi(s_j, a_j)^\top (w_t^* - w_{t+1}), \psi_{\theta_t}(a_j|s_j)\rangle\right)^\top (\lambda_t^\top \nabla J(\theta_t))\right\rangle\right]
$$

$$
+ \mathbb{E}\left[\left\langle \lambda_t - \bar{\lambda}, \left(\frac{1}{N_{\text{FC}}} \sum_{j=0}^{N_{\text{FC}}-1} \langle \phi(s_j, a_j)^\top w_{t+1}, \psi_{\theta_t}(a_j|s_j)\rangle\right)^\top \right.\right.
$$
$$
\left.\left. \cdot \left(\lambda_t^\top \nabla J(\theta_t) - \lambda_t^\top \frac{1}{N_{\text{FC}}} \sum_{l=0}^{N_{\text{FC}}-1} \langle \phi(s_l, a_l)^\top w_t^*, \psi_{\theta_t}(a_l|s_l)\rangle\right)\right\rangle\right]
$$

$$
+ \mathbb{E}\left[\left\langle \lambda_t - \bar{\lambda}, \left(\frac{1}{N_{\text{FC}}} \sum_{j=0}^{N_{\text{FC}}-1} \langle \phi(s_j, a_j)^\top w_{t+1}, \psi_{\theta_t}(a_j|s_j)\rangle\right)^\top \right.\right.
$$
$$
\left.\left. \cdot \left(\lambda_t^\top \frac{1}{N_{\text{FC}}} \sum_{l=0}^{N_{\text{FC}}-1} \langle \phi(s_l, a_l)^\top (w_t^* - w_{t+1}), \psi_{\theta_t}(a_l|s_l)\rangle\right)\right\rangle\right]
$$

$$
+ \mathbb{E}\left[\left\langle \lambda_t - \bar{\lambda}, \left(\frac{1}{N_{\text{FC}}} \sum_{j=0}^{N_{\text{FC}}-1} \langle \phi(s_j, a_j)^\top w_{t+1}, \psi_{\theta_t}(a_j|s_j)\rangle\right)^\top \right.\right.
$$
$$
\left.\left. \cdot \left(\lambda_t^\top \frac{1}{N_{\text{FC}}} \sum_{l=0}^{N_{\text{FC}}-1} \langle \phi(s_l, a_l)^\top w_{t+1}, \psi_{\theta_t}(a_l|s_l)\rangle\right)\right\rangle\right]
$$

$$
\stackrel{(i)}{\leq} \frac{C_\phi}{1-\gamma}\epsilon_{\text{app}} + \frac{C_\phi^2}{1-\gamma} \max_{k\in[K]} \mathbb{E}[\|w_t^{*k} - w_{t+1}^k\|_2] + C_\phi^3 B\epsilon_{\text{app}} + \mathbb{E}\left[\left\langle \lambda_t - \bar{\lambda}, \left(\frac{1}{N_{\text{FC}}}\right.\right.\right.
$$
$$
\left.\left.\left.\sum_{j=0}^{N_{\text{FC}}-1} \langle \phi(s_j, a_j)^\top w_{t+1}, \psi_{\theta_t}(a_j|s_j)\rangle\right)^\top \left(\lambda_t^\top \frac{1}{N_{\text{FC}}} \sum_{l=0}^{N_{\text{FC}}-1} \langle \phi(s_l, a_l)^\top w_{t+1}, \psi_{\theta_t}(a_l|s_l)\rangle\right)\right\rangle\right],
$$
$$
\tag{30}
$$

where $(i)$ follows from Assumption 4.1 and Definition 4.4. Then we consider the last term of the above inequality. We first follow the non-expansive property of projection onto the convex set

$$
\|\lambda_{t+1} - \bar{\lambda}\|_2^2
$$
$$
\leq \left\|\lambda_t - \bar{\lambda} - c_t \lambda_t^\top \left(\frac{1}{N_{\text{FC}}} \sum_{j=0}^{N_{\text{FC}}-1} \langle \phi(s_j, a_j)^\top w_{t+1}, \psi_{\theta_t}(a_j|s_j)\rangle\right) \left(\frac{1}{N_{\text{FC}}} \sum_{l=0}^{N_{\text{FC}}-1} \langle \phi(s_l, a_l)^\top w_{t+1}, \psi_{\theta_t}(a_l|s_l)\rangle\right)^\top\right\|_2^2
$$
$$
= \|\lambda_t - \bar{\lambda}\|_2^2
$$
$$
+ c_t^2 \underbrace{\left\|\lambda_t^\top \left(\frac{1}{N_{\text{FC}}} \sum_{j=0}^{N_{\text{FC}}-1} \langle \phi(s_j, a_j)^\top w_{t+1}, \psi_{\theta_t}(a_j|s_j)\rangle\right) \left(\frac{1}{N_{\text{FC}}} \sum_{l=0}^{N_{\text{FC}}-1} \langle \phi(s_l, a_l)^\top w_{t+1}\rangle \psi_{\theta_t}(a_l|s_l)\rangle\right)^\top\right\|_2^2}_{\text{term A}}
$$
$$
- 2c_t \underbrace{\left\langle \lambda_t - \bar{\lambda}, \lambda_t^\top \left(\frac{1}{N_{\text{FC}}} \sum_{j=0}^{N_{\text{FC}}-1} \langle \phi(s_j, a_j)^\top w_{t+1}, \psi_{\theta_t}(a_j|s_j)\rangle\right) \left(\frac{1}{N_{\text{FC}}} \sum_{l=0}^{N_{\text{FC}}-1} \langle \phi(s_l, a_l)^\top w_{t+1}, \psi_{\theta_t}(a_l|s_l)\rangle\right)^\top\right\rangle}_{\text{term B}}.
$$
$$
\tag{31}
$$

For term $A$, we have

term A

$$\leq c_t^2 \Big\| \lambda_t^\top \frac{1}{N_{\text{FC}}} \sum_{j=0}^{N_{\text{FC}}-1} \langle \phi(\boldsymbol{s}_j, \boldsymbol{a}_j)^\top \boldsymbol{w}_{t+1}, \psi_{\theta_t}(\boldsymbol{a}_j|\boldsymbol{s}_j) \rangle \Big\|_2^2 \Big\| \frac{1}{N_{\text{FC}}} \sum_{l=0}^{N_{\text{FC}}-1} \langle \phi(\boldsymbol{s}_l, \boldsymbol{a}_l)^\top \boldsymbol{w}_{t+1}, \psi_{\theta_t}(\boldsymbol{a}_l|\boldsymbol{s}_l) \rangle \Big\|_2^2$$

$$\overset{(i)}{\leq} c_t^2 C_\phi^2 B \Big\| \lambda_t^\top \frac{1}{N_{\text{FC}}} \sum_{j=0}^{N_{\text{FC}}-1} \langle \phi(\boldsymbol{s}_j, \boldsymbol{a}_j)^\top \boldsymbol{w}_{t+1}, \psi_{\theta_t}(\boldsymbol{a}_j|\boldsymbol{s}_j) \rangle \Big\|_2^2$$

$$\leq 2 c_t^2 C_\phi^2 B \Big\| \lambda_t^\top \frac{1}{N_{\text{FC}}} \sum_{j=0}^{N_{\text{FC}}-1} \langle \phi(\boldsymbol{s}_j, \boldsymbol{a}_j)^\top \boldsymbol{w}_t^*, \psi_{\theta_t}(\boldsymbol{a}_j|\boldsymbol{s}_j) \rangle \Big\|_2^2$$

$$+ 2 c_t^2 C_\phi^2 B \Big\| \lambda_t^\top \frac{1}{N_{\text{FC}}} \sum_{j=0}^{N_{\text{FC}}-1} \langle \phi(\boldsymbol{s}_j, \boldsymbol{a}_j)^\top (\boldsymbol{w}_{t+1} - \boldsymbol{w}_t^*), \psi_{\theta_t}(\boldsymbol{a}_j|\boldsymbol{s}_j) \rangle \Big\|_2^2, \tag{32}$$

where $(i)$ follows from Assumption 4.1. Then we take expectations on both sides,

$$\mathbb{E}[\text{term A}] \leq 2 c_t^2 C_\phi^2 B \mathbb{E}\Big[ \Big\| \lambda_t^\top \frac{1}{N_{\text{FC}}} \sum_{j=0}^{N_{\text{FC}}-1} \langle \phi(\boldsymbol{s}_j, \boldsymbol{a}_j)^\top (\boldsymbol{w}_{t+1} - \boldsymbol{w}_t^*), \psi_{\theta_t}(\boldsymbol{a}_j|\boldsymbol{s}_j) \rangle \Big\|_2^2 \Big]$$

$$+ 4 c_t^2 C_\phi^2 B \mathbb{E}\Big[ \Big\| \lambda_t^\top \frac{1}{N_{\text{FC}}} \sum_{j=0}^{N_{\text{FC}}-1} \langle (\phi(\boldsymbol{s}_j, \boldsymbol{a}_j)^\top \boldsymbol{w}_t^*) - Q_{\theta_t}(\boldsymbol{s}_j, \boldsymbol{a}_j)), \psi_{\theta_t}(\boldsymbol{a}_j|\boldsymbol{s}_j) \rangle \Big\|_2^2 \Big]$$

$$+ 4 c_t^2 C_\phi^2 B \mathbb{E}\Big[ \Big\| \lambda_t^\top \frac{1}{N_{\text{FC}}} \sum_{j=0}^{N_{\text{FC}}-1} Q_{\theta_t}(\boldsymbol{s}_j, \boldsymbol{a}_j) \psi_{\theta_t}(\boldsymbol{a}_j|\boldsymbol{s}_j) \Big\|_2^2 - \| \lambda_t^\top \nabla J(\theta_t) \|_2^2 \Big]$$

$$+ 4 c_t^2 C_\phi^2 B \mathbb{E}[\| \lambda_t^\top \nabla J(\theta_t) \|_2^2]$$

$$\overset{(i)}{\leq} 2 c_t^2 C_\phi^4 B \max_{k \in [K]} \mathbb{E}[\| w_{t+1}^k - w_t^{*k} \|_2^2] + 4 c_t^2 C_\phi^4 B \mathbb{E}[\| \langle \phi(\boldsymbol{s}_j, \boldsymbol{a}_j), \boldsymbol{w}_t^* \rangle - Q_{\theta_t}(\boldsymbol{s}_j, \boldsymbol{a}_j) \|_2^2]$$

$$+ \frac{4 c_t^2 C_\phi^8 B^4}{N_{\text{FC}}} + 4 c_t^2 C_\phi^2 B \mathbb{E}[\| \lambda_t^\top \mathbb{E}_{d_{\theta_t}} [\boldsymbol{\zeta}(\boldsymbol{s}, \boldsymbol{a}, \theta_t, \boldsymbol{w}_{t+1})] \|_2^2]$$

$$\overset{(ii)}{\leq} 2 c_t^2 C_\phi^4 B \left( \frac{4 B^2}{N_{\text{critic}} + 1} + \frac{U_\delta^2 C_\phi^2 \log N_{\text{critic}}}{4 \lambda_A^2 (N_{\text{critic}} + 1)} \right) + 4 c_t^2 C_\phi^2 B \epsilon_{\text{app}}^2 + \frac{4 c_t^2 C_\phi^8 B^4}{N_{\text{FC}}}$$

$$+ 4 c_t^2 C_\phi^2 B \mathbb{E}[\| \lambda_t^\top \mathbb{E}_{d_{\theta_t}} [\langle \phi(\boldsymbol{s}, \boldsymbol{a})^\top \boldsymbol{w}_{t+1}, \psi_{\theta_t}(\boldsymbol{s}, \boldsymbol{a}) \rangle] \|_2^2],$$

where $(i)$ follows from Assumption 4.1 and Xu & Gu (2020) and $(ii)$ follows from Lemma B.1 and Definition 4.4. Then for term $B$, we have

$$\mathbb{E}[\text{term B}]$$

$$= 2 c_t \mathbb{E}\Big[ \langle \lambda_t - \bar{\lambda}, \Big( \frac{1}{N_{\text{FC}}} \sum_{j=0}^{N_{\text{FC}}-1} \langle \phi(\boldsymbol{s}_j, \boldsymbol{a}_j)^\top \boldsymbol{w}_{t+1}, \psi_{\theta_t}(\boldsymbol{a}_j|\boldsymbol{s}_j) \rangle \Big)^\top$$

$$\cdot \Big( \lambda_t^\top \frac{1}{N_{\text{FC}}} \sum_{l=0}^{N_{\text{FC}}-1} \langle \phi(\boldsymbol{s}_l, \boldsymbol{a}_l)^\top \boldsymbol{w}_{t+1}, \psi_{\theta_t}(\boldsymbol{a}_l|\boldsymbol{s}_l) \rangle \Big) \Big]$$

$$\leq \mathbb{E}[\| \lambda_t - \bar{\lambda} \|_2^2 - \| \lambda_{t+1} - \bar{\lambda} \|_2^2] + 2 c_t^2 C_\phi^4 B \left( \frac{4 B^2}{N_{\text{critic}} + 1} + \frac{U_\delta^2 C_\phi^2 \log N_{\text{critic}}}{4 \lambda_A^2 (N_{\text{critic}} + 1)} \right) + 4 c_t^2 C_\phi^2 B \epsilon_{\text{app}}^2$$

$$+ \frac{4 c_t^2 C_\phi^8 B^4}{N_{\text{FC}}} + 4 c_t^2 C_\phi^2 B \mathbb{E}\left[ \| \lambda_t^\top \mathbb{E}_{d_{\theta_t}} [\langle \phi(\boldsymbol{s}, \boldsymbol{a})^\top \boldsymbol{w}_{t+1}, \psi_{\theta_t}(\boldsymbol{s}, \boldsymbol{a}) \rangle] \|_2^2 \right]. \tag{33}$$

Then we substitute Equation (33) into Equation (30), we can derive:

$$\beta_t \text{term II} = \beta_t \mathbb{E}[\langle \lambda_t - \bar{\lambda}, (\nabla J(\theta_t))^\top (\lambda_t^\top \nabla J(\theta_t)) \rangle]$$

$$\leq \beta_t \frac{C_\phi}{1-\gamma}\epsilon_{\text{app}} + \beta_t \frac{C_\phi^2}{1-\gamma}\max_{k\in[K]}\mathbb{E}[\|w_t^{*k} - w_{t+1}^k\|_2] + \frac{\beta_t}{2c_t}\mathbb{E}[\|\lambda_t - \bar{\lambda}\|_2^2 - \|\lambda_{t+1} - \bar{\lambda}\|_2^2]$$

$$+ \beta_t c_t C_\phi^4 B\left(\frac{4B^2}{N_{\text{critic}}+1} + \frac{U_\delta^2 C_\phi^2 \log N_{\text{critic}}}{4\lambda_A^2(N_{\text{critic}}+1)}\right) + 2\beta_t c_t C_\phi^2 B\epsilon_{\text{app}}^2 + \beta_t C_\phi^3 B\epsilon_{\text{app}} + \frac{2\beta_t c_t C_\phi^8 B^4}{N_{\text{FC}}}$$

$$+ 2\beta_t c_t C_\phi^2 B\mathbb{E}[\|\lambda_t^\top \mathbb{E}_{d_{\theta_t}}\left[\langle\phi(\boldsymbol{s},\boldsymbol{a})^\top \boldsymbol{w}_{t+1}, \psi_{\theta_t}(\boldsymbol{s},\boldsymbol{a})\rangle\right]\|_2^2]. \tag{34}$$

Plug Equation (29) and Equation (34) in Equation (28), we can get that

$$\beta_t\mathbb{E}[\|\lambda_t^\top \nabla J(\theta_t)\|_2^2]$$

$$\leq \mathbb{E}[\bar{\lambda}^\top J(\theta_{t+1})] - \mathbb{E}[\bar{\lambda}^\top J(\theta_t)] + \beta_t\text{term I} + \beta_t\text{term II} + \beta_t^2 \frac{L_J C_\phi^4 B^2}{N_{\text{actor}}}$$

$$+ \frac{L_J\beta_t^2}{2}\|\lambda_t^\top \mathbb{E}_{d_{\theta_t}}[\langle\phi(\boldsymbol{s},\boldsymbol{a})^\top \boldsymbol{w}_{t+1}, \psi_{\theta_t}(\boldsymbol{s},\boldsymbol{a})\rangle]\|_2^2 + \frac{C_\phi^2\beta_t}{1-\gamma}\left(\sqrt{\frac{4B^2}{N_{\text{critic}}+1} + \frac{U_\delta^2 C_\phi^2 \log N_{\text{critic}}}{4\lambda_A^2(N_{\text{critic}}+1)}}\right)$$

$$\leq \mathbb{E}[\bar{\lambda}J(\theta_{t+1})] - \mathbb{E}[\bar{\lambda}J(\theta_t)] + \beta_t\left(\frac{C_\phi^2}{1-\gamma} + \frac{C_\phi}{1-\gamma} + C_\phi^3 B + 2c_t C_\phi^2 B\epsilon_{\text{app}}\right)\epsilon_{\text{app}}$$

$$+ \frac{\beta_t}{2c_t}\mathbb{E}[\|\lambda_t - \bar{\lambda}\|_2^2 - \|\lambda_{t+1} - \bar{\lambda}\|_2^2] + c_t C_\phi^4 B\beta_t\left(\frac{4B^2}{N_{\text{critic}}+1} + \frac{U_\delta^2 C_\phi^2 \log N_{\text{critic}}}{4\lambda_A^2(N_{\text{critic}}+1)}\right)$$

$$+ \frac{C_\phi^2\beta_t}{1-\gamma}\sqrt{\frac{4B^2}{N_{\text{critic}}+1} + \frac{U_\delta^2 C_\phi^2 \log N_{\text{critic}}}{4\lambda_A^2(N_{\text{critic}}+1)}} + \frac{2C_\phi^6 B^4 \beta_t c_t}{N_{\text{FC}}} + \frac{C_\phi^4 B^2 L_J\beta_t^2}{N_{\text{actor}}}$$

$$+ \left(\frac{L_J\beta_t^2}{2} + 2\beta_t c_t C_\phi^2 B\right)\mathbb{E}[\|\lambda_t^\top \mathbb{E}_{d_{\theta_t}}\left[\langle\phi(\boldsymbol{s},\boldsymbol{a})^\top \boldsymbol{w}_{t+1}, \psi_{\theta_t}(\boldsymbol{s},\boldsymbol{a})\rangle\right]\|_2^2]. \tag{35}$$

Next, we consider the bound between $\left\|\lambda_t^\top \mathbb{E}_{d_{\theta_t}}\left[\langle\phi(\boldsymbol{s},\boldsymbol{a})^\top \boldsymbol{w}_{t+1}, \psi_{\theta_t}(\boldsymbol{s},\boldsymbol{a})\rangle\right]\right\|_2^2 - \left\|\lambda_t^\top \nabla J(\theta_t)\right\|_2^2$:

$$\left\|\lambda_t^\top \mathbb{E}_{d_{\theta_t}}\left[\langle\phi(\boldsymbol{s},\boldsymbol{a})^\top \boldsymbol{w}_{t+1}, \psi_{\theta_t}(\boldsymbol{s},\boldsymbol{a})\rangle\right]\right\|_2 - \left\|\lambda_t^\top \nabla J(\theta_t)\right\|_2$$

$$= \left\|\lambda_t^\top \mathbb{E}_{d_{\theta_t}}\left[\langle\phi(\boldsymbol{s},\boldsymbol{a})^\top \boldsymbol{w}_{t+1}, \psi_{\theta_t}(\boldsymbol{s},\boldsymbol{a})\rangle\right]\right\|_2 - \left\|\lambda_t^\top \mathbb{E}_{d_{\theta_t}}\left[\langle\phi(\boldsymbol{s},\boldsymbol{a})^\top \boldsymbol{w}_t^*, \psi_{\theta_t}(\boldsymbol{s},\boldsymbol{a})\rangle\right]\right\|_2$$

$$+ \left\|\lambda_t^\top \mathbb{E}_{d_{\theta_t}}\left[\langle\phi(\boldsymbol{s},\boldsymbol{a})^\top \boldsymbol{w}_t^*, \psi_{\theta_t}(\boldsymbol{s},\boldsymbol{a})\rangle\right]\right\|_2 - \left\|\lambda_t^\top \nabla J(\theta_t)\right\|_2$$

$$\leq \left\|\lambda_t^\top \mathbb{E}_{d_{\theta_t}}\left[\langle\phi(\boldsymbol{s},\boldsymbol{a})^\top (\boldsymbol{w}_t^* - \boldsymbol{w}_{t+1}), \psi_{\theta_t}(\boldsymbol{s},\boldsymbol{a})\rangle\right]\right\|_2$$

$$+ \left\|\lambda_t^\top \mathbb{E}_{d_{\theta_t}}\left[\langle Q_{\theta_t}(\boldsymbol{s},\boldsymbol{a}) - \phi(\boldsymbol{s},\boldsymbol{a})^\top \boldsymbol{w}_t^*, \psi_{\theta_t}(\boldsymbol{s},\boldsymbol{a})\rangle\right]\right\|_2$$

$$\overset{(i)}{\leq} \max_k\left\{\left\|\phi^k(s^k,a^k)\right\|_2 \left\|w_t^{*k} - w_{t+1}^k\right\|_2 \left\|\psi_{\theta_t}(s^k,a^k)\right\|_2\right\}$$

$$+ \max_k\left\{\sqrt{E_{d_{\theta_t}}\left[\left\|Q_{\theta_t}^k(s_k,a_k) - \phi^\top(s_k,a_k)w_t^{*k}\right\|_2^2\right]}\left\|\psi_{\theta_t}(s^k,a^k)\right\|_2\right\}$$

$$\overset{(ii)}{\leq} C_\phi^2\left\|w_t^{*k} - w_{t+1}^k\right\|_2 + C_\phi\epsilon_{\text{app}}, \tag{36}$$

where $(i)$ follows from Cauchy-Schwarz inequaltiy and $(ii)$ follows from Definition 4.4. Then, we can get that

$$\mathbb{E}\left[\left\|\lambda_t^\top \mathbb{E}_{d_{\theta_t}}\left[\langle\phi(\boldsymbol{s},\boldsymbol{a})^\top \boldsymbol{w}_{t+1}, \psi_{\theta_t}(\boldsymbol{s},\boldsymbol{a})\rangle\right]\right\|_2^2 - \left\|\lambda_t^\top \nabla J(\theta_t)\right\|_2^2\right]$$

$$\leq \mathbb{E}\bigg[\left(\left\|\lambda_t^\top \mathbb{E}_{d_{\theta_t}}\left[\langle\phi(\boldsymbol{s},\boldsymbol{a})^\top \boldsymbol{w}_{t+1}, \psi_{\theta_t}(\boldsymbol{s},\boldsymbol{a})\rangle\right]\right\|_2 - \left\|\lambda_t^\top \nabla J(\theta_t)\right\|_2\right)$$

$$\times \left(\left\|\lambda_t^\top \mathbb{E}_{d_{\theta_t}}\left[\langle\phi(\boldsymbol{s},\boldsymbol{a})^\top \boldsymbol{w}_{t+1}, \psi_{\theta_t}(\boldsymbol{s},\boldsymbol{a})\rangle\right]\right\|_2 + \left\|\lambda_t^\top \nabla J(\theta_t)\right\|_2\right)\bigg]$$

$$\overset{(i)}{\leq} \left(C_\phi^2 B + \frac{C_\phi}{1-\gamma}\right)\left(C_\phi^2\mathbb{E}\left[\left\|w_t^{*k} - w_{t+1}^k\right\|_2\right] + C_\phi\epsilon_{\text{app}}\right)$$

$$\overset{(ii)}{\leq} \left( C_\phi^4 B + \frac{C_\phi^3}{1-\gamma} \right) \sqrt{ \frac{4B^2}{N_{\text{critic}}+1} + \frac{U_\delta^2 C_\phi^2 \log N_{\text{critic}}}{4\lambda_A^2 (N_{\text{critic}}+1)} } + \left( C_\phi^3 B + \frac{C_\phi^2}{1-\gamma} \right) \epsilon_{\text{app}}, \quad (37)$$

where $(i)$ follows from Definition 4.4. We substitute Equation (37) into Equation (35),

$$\left( \beta_t - \frac{L_J \beta_t^2}{2} - 2\beta_t c_t C_\phi^2 B \right) \mathbb{E}[\|\lambda_t^\top \nabla J(\theta_t)\|_2^2]$$

$$\leq \mathbb{E}[\bar{\lambda}^\top J(\theta_{t+1})] - \mathbb{E}[\bar{\lambda}^\top J(\theta_t)] + \frac{\beta_t}{2c_t} \mathbb{E}[\|\lambda_t - \bar{\lambda}\|_2^2 - \|\lambda_{t+1} - \bar{\lambda}\|_2^2]$$

$$+ \beta_t \left( \left( \frac{L_J \beta_t}{2} + 2c_t C_\phi^2 B \right) \left( C_\phi^3 B + \frac{C_\phi^2}{1-\gamma} \right) + \frac{C_\phi^2}{1-\gamma} + \frac{C_\phi}{1-\gamma} + C_\phi^3 B + 2c_t C_\phi^2 B \epsilon_{\text{app}} \right) \epsilon_{\text{app}}$$

$$+ \beta_t \left( \left( \frac{L_J \beta_t}{2} + 2c_t C_\phi^2 B \right) \left( C_\phi^4 B + \frac{C_\phi^3}{1-\gamma} \right) + \frac{C_\phi^2}{1-\gamma} \right) \sqrt{ \frac{4B^2}{N_{\text{critic}}+1} + \frac{U_\delta^2 C_\phi^2 \log N_{\text{critic}}}{4\lambda_A^2 (N_{\text{critic}}+1)} }$$

$$+ c_t \beta_t C_\phi^4 B \left( \frac{4B^2}{N_{\text{critic}}+1} + \frac{U_\delta^2 C_\phi^2 \log N_{\text{critic}}}{4\lambda_A^2 (N_{\text{critic}}+1)} \right) + \frac{2C_\phi^6 B^4 \beta_t c_t}{N_{\text{FC}}} + \frac{C_\phi^4 B^2 L_J \beta_t^2}{N_{\text{actor}}}.$$

Since we choose $\beta_t = \beta \leq \frac{1}{L_J}$, $c_t = c' \leq \frac{1}{8 C_\phi^2 B}$, we can guarantee that $\frac{\beta_t}{2} - \frac{\beta_t^2}{2} - 4c_t \beta_t C_\phi^2 B \geq \frac{\beta}{4}$. Then, by rearranging the above inequality, we can have

$$\frac{\beta}{4} \mathbb{E}[\|\lambda_t^\top \nabla J(\theta_t)\|_2^2] \leq \mathbb{E}[\bar{\lambda}^\top J(\theta_{t+1})] - \mathbb{E}[\bar{\lambda}^\top J(\theta_t)] + \frac{\beta}{2c'} \mathbb{E}[\|\lambda_t - \bar{\lambda}\|_2^2 - \|\lambda_{t+1} - \bar{\lambda}\|_2^2]$$

$$+ \beta \left( \left( \frac{L_J \beta}{2} + 2c' C_\phi^2 B \right) \left( C_\phi^3 B + \frac{C_\phi^2}{1-\gamma} \right) + \frac{C_\phi^2}{1-\gamma} + \frac{C_\phi}{1-\gamma} + C_\phi^3 B + 2c' C_\phi^2 B \epsilon_{\text{app}} \right) \epsilon_{\text{app}}$$

$$+ \beta \left( \left( \frac{L_J \beta}{2} + 2c' C_\phi^2 B \right) \left( C_\phi^4 B + \frac{C_\phi^3}{1-\gamma} \right) + \frac{C_\phi^2}{1-\gamma} \right) \sqrt{ \frac{4B^2}{N_{\text{critic}}+1} + \frac{U_\delta^2 C_\phi^2 \log N_{\text{critic}}}{4\lambda_A^2 (N_{\text{critic}}+1)} }$$

$$+ \beta c' C_\phi^4 B \left( \frac{4B^2}{N_{\text{critic}}+1} + \frac{U_\delta^2 C_\phi^2 \log N_{\text{critic}}}{4\lambda_A^2 (N_{\text{critic}}+1)} \right) + \frac{2C_\phi^6 B^4 \beta c'}{N_{\text{FC}}} + \frac{C_\phi^4 B^2 L_J \beta^2}{N_{\text{actor}}}.$$

Then, telescoping over $t = 0, 1, 2, ..., T-1$ yields,

$$\frac{1}{T} \sum_{t=0}^{T-1} \mathbb{E}[\|\lambda_t^\top \nabla J(\theta_t)\|_2^2] \leq \frac{4}{\beta T} \mathbb{E}[\bar{\lambda}^\top J(\theta_T) - \bar{\lambda}^\top J(\theta_0)] + \frac{2}{c'T} \mathbb{E}[\|\lambda_0 - \bar{\lambda}\|_2^2 - \|\lambda_T - \bar{\lambda}\|_2^2]$$

$$+ 4 \left( \left( \frac{L_J \beta}{2} + 2c' C_\phi^2 B \right) \left( C_\phi^3 B + \frac{C_\phi^2}{1-\gamma} \right) + \frac{C_\phi^2}{1-\gamma} + \frac{C_\phi}{1-\gamma} + C_\phi^3 B + 2c' C_\phi^2 B \epsilon_{\text{app}} \right) \epsilon_{\text{app}}$$

$$+ 4 \left( \left( \frac{L_J \beta}{2} + 2c' C_\phi^2 B \right) \left( C_\phi^4 B + \frac{C_\phi^3}{1-\gamma} \right) + \frac{C_\phi^2}{1-\gamma} \right) \sqrt{ \frac{4B^2}{N_{\text{critic}}+1} + \frac{U_\delta^2 C_\phi^2 \log N_{\text{critic}}}{4\lambda_A^2 (N_{\text{critic}}+1)} }$$

$$+ 4c' C_\phi^4 B \left( \frac{4B^2}{N_{\text{critic}}+1} + \frac{U_\delta^2 C_\phi^2 \log N_{\text{critic}}}{4\lambda_A^2 (N_{\text{critic}}+1)} \right) + \frac{8C_\phi^6 B^4 c'}{N_{\text{FC}}} + \frac{4C_\phi^4 B^2 L_J \beta}{N_{\text{actor}}}.$$

Lastly, since $\lambda_t^* = \arg\min_{\lambda \in \Lambda} \|\lambda^\top \nabla J(\theta_t)\|_2^2$, we have

$$\frac{1}{T} \sum_{t=0}^{T-1} \mathbb{E}[\|(\lambda_t^*)^\top \nabla J(\theta_t)\|_2^2] \leq \frac{1}{T} \sum_{t=0}^{T-1} \mathbb{E}[\|\lambda_t^\top \nabla J(\theta_t)\|_2^2]$$

$$= \mathcal{O}\left( \frac{1}{\beta T} + \frac{1}{c'T} + \epsilon_{\text{app}} + \frac{1}{\sqrt{N_{\text{critic}}}} + \frac{\beta}{N_{\text{actor}}} + \frac{c'}{N_{\text{FC}}} \right).$$

The proof is complete.

### D.2 PROOF OF COROLLARY 4.9

Since we choose $\beta = \mathcal{O}(1)$ and $c' = \mathcal{O}(1)$, we have

$$\frac{1}{T} \sum_{t=0}^{T-1} \mathbb{E}[\|(\lambda_t^*)^\top \nabla J(\theta_t)\|_2^2] = \mathcal{O}\Big(\frac{1}{T} + \frac{1}{\sqrt{N_{\text{critic}}}} + \frac{1}{N_{\text{actor}}} + \frac{1}{N_{\text{FC}}} + \epsilon_{\text{app}}\Big).$$

To achieve an $\epsilon$-accurate Pareto stationary policy, it requires $T = \mathcal{O}(\epsilon^{-1})$, $N_{\text{critic}} = \mathcal{O}(\epsilon^{-2})$, $N_{\text{actor}} = \mathcal{O}(\epsilon^{-1})$, $N_{\text{FC}} = \mathcal{O}(\epsilon^{-1})$, and each objective requires $\mathcal{O}(\epsilon^{-3})$ samples. $\qquad\square$

