# OpenReview forum: "Finite-Time Analysis for Conflict-Avoidant Multi-Task Reinforcement Learning"
_ICLR.cc/2025/Conference — ICLR 2025 Conference Withdrawn Submission_

### Official Review · Reviewer_Gp19 · 2024-10-16

**Soundness:** 3
**Presentation:** 3
**Contribution:** 2
**Rating:** 5
**Confidence:** 3

**Summary:**

The paper studies the multi-task reinforcement learning problem from a theoretical perspective. Different from many of the existing literature, the paper focus on addressing the issue of gradient conflict brought by the imbalanced gradients from different tasks. In this paper, the author proposes a formulation that is able to dynamically weight the task importance, using two updates, named as CA and FC. For the convergence results, the paper demonstrates that the proposed MTAC-CA method enjoys a sample complexity of $O(\epsilon^-5)$ and MTAC-FC method has a complexity of $O(\epsilon^-3)$ to find a pareto stationary policy, up to function approximation errors. The paper also uses numerical examples to demonstrate the effectiveness of the algorithm.

**Strengths:**

* The paper studies a theoretically sound problem, and the theoretical results derived look correct from my brief examination.
* The paper structure is complete, and the presentation is good. The problem formulation and the results in the paper are easy to follow.
* I appreciated that the author has numerical experiments in the paper and also provides the code for the experiment, which are sometimes not the case for many RL theory paper.

**Weaknesses:**

I mainly have the following two concerns
* The analysis for the TD learning part looks standard. Yet it seems neither did the author claim this analysis as original contribution nor did the author cite any reference in the proof (please point me to the place if I missed that).
* The overall novelty of the analysis remains questionable. The research problem sounds good to me. However, given (Xiao et al., 2023), which studies the general multi-task learning problem, I am not sure how much does this paper contributes to the community. If the contribution is only about the function approximation/gradient estimation from the RL side, I feel like this adds-on is not enough for ICLR.

**Questions:**

I hope the author could provide further clarifications on the two points that I mentioned in the weakness section. If they can be addressed, I  will consider adjusting my score.

---

> ### Author Response · Authors · 2024-11-21
>
> ***Q1: The analysis for the TD learning part should be supported with appropriate references***
>
> Thanks for the comments. We agree that the TD analysis part is quite standard, however, our major contribution lies in the other analysis part (discussed below). We have modified the TD part to make this more clear.
>
> ***Q2: The overall novelty of the analysis remains questionable***
>
> Thank you for the question. We face two main technical difficulties in our analysis. The first difficulty lies in addressing gradient estimation bias, as MOO analyses typically rely on **unbiased** gradient estimators. However, in our setting, the function approximation introduces a non-vanishing gradient estimation error.
>
> To elaborate, in **supervised learning**, **unbiased** gradient estimators for each task can be easily obtained using a double sampling strategy as shown in (Chen et.al., 2023), (Xiao et.al., 2022). Although previous work (Fernandoetal et.al., 2022) introduced a tracking variable to manage this bias, it requires the bias to vanish over iterations. In contrast, in **multi-task reinforcement learning**, obtaining an unbiased gradient estimator for both task gradient and weight gradient is impossible due to function approximation error in the critic. These non-vanishing errors introduce additional bias, as the CA direction is updated by biased estimated policy gradients. Bounding this extra error remains an open question in RL and MOO problems.
>
> The second difficulty arises in bounding the CA distance. Due to the non-vanishing error, directly bounding this CA distance (see line 446) leads to non-vanishing error terms. Instead, we introduce a surrogate direction and decompose the error into three error components. The CA distance can then be bounded by the gap between $\lambda_{t,N_{\textbf{CA}}}$ and $\widehat{\lambda}_t^\prime$, plus the gap between $\widehat{\lambda}_t^\prime$ and $\lambda_t^*$. We also refer the reviewer to Section 5 for a more detailed discussion.

---

### Official Review · Reviewer_ZxAe · 2024-10-30

**Soundness:** 3
**Presentation:** 3
**Contribution:** 2
**Rating:** 3
**Confidence:** 2

**Summary:**

The paper presents a solution to Multi-Task RL, where we wish to solve more than one task with the same policy. In such cases, it is possible that the gradients we obtain from different tasks are conflicting and as such we would like to ensure we find a Pareto optimal solution. The paper proposes two algorithms for this problem setting, one that focuses on fast convergence and the other that keeps the CA-distance (a measure of disagreement between different tasks' gradients). The authors also present empirical results on the MT10 benchmark.

**Strengths:**

- The paper is generally well-written, easy to follow and focuses on important problem in the RL community

**Weaknesses:**

- It seems like the ideas for the proposed algorithms are borrowed directly from Xiao et al. (2023). The only novelty seems to be the explicitly obtain the bounds for when the previously developed algorithms are applied to Actor-Critic style RL. However, I believe the convergence of Actor-Critic algorithms has been extensively studied before, e.g. Barakat et al. (2022). As such, I do not see much novelty in the proposed approach. It seems like a simple combination of two existing analyses with no added value (an A + B type of paper).
- The SDMGrad algorithm proposed by Xiao et al. (2023) has been applied to RL problems and in fact, one of the benchmark problems they present is MT10, also considered by the authors of this submission. However, for some reason, the authors do not compare it with SDMGrad. If we, however, look at the values reported by Xiao et al. (2023), it seems like SDMGrad achieves better values than the algorithm presented in the submission. As such, I believe there is also no empirical novelty, as the proposed algorithm performs worse than existing algorithms.

**Questions:**

- Can authors clarify what are the exact differences compared to the work of Xiao et al. (2023)? Are there any unique challenges, when using their results in the RL setting? Is there anything beyond a simple A + B type combination of existing results?
- Why is the previous, most similar work of SDMGrad not shown in the experimental evaluation?

---

> ### Author Response · Authors · 2024-11-21
>
> ***Q1 \& Weaknesses 1: Are there any unique challenges, when using their results in the RL setting?***
>
> Thanks for the comment. We are aware of the techniques to handle the MOO problem with unbiased gradient estimator in (Xiao et.al., 2023) and with biased gradient estimator where the bias vanishes with the number of iterations in (Fernandoetal et. al., 2022). Nevertheless, unlike the {\bf vanishingly biased} or {\bf unbiased} gradient estimator in the supervised case, the bias of policy gradient estimation is {\bf non-vanishing} in our problem. Thus, the analysis in (Xiao et.al., 2023) can not be applied to our work directly.  With a non-vanishing biased gradient estimator, analyzing the CA distance (i.e., the distance between the real CA direction and the estimated CA direction) becomes significantly more challenging. To address this, we propose a {\bf surrogate CA direction} to effectively bound the CA distance. In addition, our analysis derives a sample complexity of $O(\epsilon^{-5})$, which is {\bf lower} than $O(\epsilon^{-6})$ in the newest version of (Xiao et. al., 2023) with small CA distance. Besides, we need to note that our analysis technique can be extended to (Xiao et. al., 2023) easily for settings settings involving non-vanishing gradient bias. This highlights that our contributions include both significant new developments and differences from the work in (Xiao et. al., 2023).
>
> ***Q2 \& Weaknesses 2: Why SDMGrad not shown in the experimental evaluation?***
>
> We would like to emphasize that our work focuses on addressing the challenge of ***non-vanishing bias*** in the gradient estimator within the multi-task reinforcement learning (MTRL) setting. Adding a regularization term like SDMGrad does not solve this issue, and hence its performance is expected to be similar to the baseline we used.  The regularization term can be similarly incorporated in our algorithm, but it is orthogonal to our contribution.  Besides, adding a regularization term may introduce confusion and detract from the clarity of our technical contributions.

---

### Official Review · Reviewer_wtS3 · 2024-11-02

**Soundness:** 2
**Presentation:** 3
**Contribution:** 2
**Rating:** 5
**Confidence:** 3

**Summary:**

The paper studies a multi-task RL problem where there is no prior weighting for each task. The paper proposes an algorithm class (with variants in dynamic weighting method) for finding a Pareto stationary policy and establishes the sample complexity. Illustrative numerical simulations are presented which show the superior performance of the proposed algorithm on an averaged multi-task objective.

**Strengths:**

The paper is well written with the background intro, algorithm development, and proof sketch clearly presented. The problem being studied is well motivated and important -- it is natural to consider multi-task RL where we do not in advance know how much weight to put on each task. The algorithms proposed are novel, with rigorous complexity analysis. I did not go through the complete proof but find the main claim and argument credible. Despite the simplicity, the simulation results do verify the superior algorithm performance.

**Weaknesses:**

1) As the weight is not given a priori, the optimization objective is very unclear. Eq.(1) is obviously not an valid objective since it is multi-dimensional. The theoretical results establish convergence to a Pareto stationary policy, defined as a point minimizing $\mathbb{E}[\min_{\lambda}||\lambda^{\top}\nabla J(\pi)||^2]$. However, this does not seem a valid metric for multi-task learning, either, as it can be made small as long as any of the tasks is well solved. In the simulations, Table 1 compares the algorithm on the averaged success rate. I suppose an averaging over all tasks is made here with equal weight for each task. This is certainly a valid objective. The issue is that, if the objective is known to be equally weighted, it defeats the purposes of dynamically adjusting the weights. In this sense, the paper fails to evaluate the proposed MTAC-CA and MTAC-FC algorithms either theoretically or in simulation.

2) The technical innovation of the work is unclear. While the paper claims "addressing this question is not easy, primarily due to the difficulty in conducting sample complexity analysis for dynamic weighting MTRL algorithms" in line 065-066, what actually makes the analysis challenging should be concretely discussed (mathematically). The analysis of actor-critic algorithms on single-task RL is well studied and this paper essentially follows the existing analysis to treat the policy and critic iterates. On the other hand, the dynamic weight adjusting schemes seem to have been studied in the CA multi-task supervised learning setting. Does adapting the analysis from supervised learning to reinforcement learning create significant challenges here? If so, the authors should highlight such challenges and how they are overcome.

**Questions:**

See weaknesses above.

---

> ### Author Response · Authors · 2024-11-21
>
> ***Weaknesses 1:***
>
> Thank you for the questions. We would answer them accordingly.
>
> (1). Eq. (1) represents a standard formulation commonly used in multi-objective optimization (MOO) or multi-task learning (MTL) frameworks, such as MoCo, MoDo, and SDMGrad. In our case, different objectives have shared policy $\pi$, yielding multiple reward functions. The goal of the MTRL problem is to simultaneously maximize these reward functions.
>
> (2). A Pareto stationary policy (or point) is a widely accepted convergence measure in MTL and MOO. Since different objectives rarely share the same optimal point, the aim is to find Pareto points where no objective can be improved without sacrificing another. The point you mentioned is indeed on the Pareto front but does not represent a balanced solution. Fixed-preference MTRL methods, such as averaging objectives, tend to converge to such points if one objective is easy to optimize. To address this, we focus on dynamic-preference MTRL methods, which aim to mitigate gradient conflicts and avoid unbalanced results by searching for the conflict-avoidant (CA) direction.
>
> (3). In our simulation, we report the average success rate but this does not mean that different objectives are updated equally. The objective is not to optimize the average rewards but to maximize multiple reward functions in a balanced way.
>
> (4). Lastly, even though objective functions can be equally weighted, it is a special case of dynamic weights that weights are all the same along the training process. Furthermore, in most cases, we have no prior knowledge of the objective preferences. Updating objectives with fixed weights will lead to gradient conflict and undesirable direction.
>
> ***Weaknesses 2:***
>
>  We face two main technical difficulties in our analysis. The first difficulty lies in addressing gradient estimation bias, as MOO analyses typically rely on **unbiased** gradient estimators. However, in our setting, the function approximation introduces a non-vanishing gradient estimation error.
>
> To elaborate, in **supervised learning**, **unbiased** gradient estimators for each task can be easily obtained using a double sampling strategy as shown in (Chen et.al., 2023), (Xiao et.al., 2022). Although previous work (Fernandoetal et.al., 2022) introduced a tracking variable to manage this bias, it requires the bias to vanish over iterations. In contrast, in **multi-task reinforcement learning**, obtaining an unbiased gradient estimator for both task gradient and weight gradient is impossible due to function approximation error in the critic. These ***non-vanishing*** errors introduce additional bias, as the CA direction is updated by biased estimated policy gradients. Bounding this extra error remains an open question in RL and MOO problems.
>
> The second difficulty arises in bounding the CA distance. Due to the non-vanishing error, directly bounding this CA distance (see line 446) leads to non-vanishing error terms. Instead, we introduce a surrogate direction and decompose the error into three error components. The CA distance can then be bounded by the gap between $\lambda_{t,N_{\textbf{CA}}}$ and $\widehat{\lambda}_t^\prime$, plus the gap between $\widehat{\lambda}_t^\prime$ and $\lambda_t^*$. Full details can be found in Appendix A.

---

> ### Comment · Reviewer_wtS3 · 2024-11-21
>
> I thank the authors for the response. Regarding your response to weakness 1.1-1.3 -- I completely agree that you should and would like to show that the proposed algorithm performs well not just on the averaged objective or on solving any single task. That is exactly where the problem lies in. The theoretical and empirical results do not show what you would like to show. I understand that Pareto stationary point may be a common metric in the literature, but it really does not provide much information on the quality of a MTRL policy if you can essentially hack the metric by solving a single task.

---

### Note · Authors · 2024-12-11

I have read and agree with the venue's withdrawal policy on behalf of myself and my co-authors.